# The Dsc ubiquitin ligase complex identifies transmembrane degrons to degrade orphaned proteins at the Golgi

Yannick Weyer [1,2,8], Sinead I. Schwabl[1,2,8], Xuechen Tang[3,8], Astha Purwar[1,2], Konstantin Siegmann[1], Angela Ruepp [1], Theresia Dunzendorfer-Matt [1], Michael A. Widerin[2], Veronika Niedrist[2], Noa J. M. Mutsters [2], Maria G. Tettamanti[4,5], Sabine Weys[2,6], Bettina Sarg[7], Leopold Kremser[7], Klaus R. Liedl [7], Oliver Schmidt [2] & David Teis[1,2] ✉

The Golgi apparatus is essential for protein sorting, yet its quality control mechanisms are poorly understood. Here we show that the Dsc ubiquitin ligase complex uses its rhomboid pseudo-protease subunit, Dsc2, to assess the hydrophobic length of α-helical transmembrane domains (TMDs) at the Golgi. Thereby the Dsc complex likely interacts with orphaned ER and Golgi proteins that have shorter TMDs and ubiquitinates them for targeted degradation. Some Dsc substrates will be extracted by Cdc48 for *e*ndosome and *G*olgi *a*ssociated proteasomal *d*egradation (EGAD), while others will undergo ESCRT dependent vacuolar degradation. Some substrates are degraded by both, EGAD- or ESCRT pathways. The accumulation of Dsc substrates entails a specific increase in glycerophospholipids with shorter and asymmetric fatty acyl chains. Hence, the Dsc complex mediates the selective degradation of orphaned proteins at the sorting center of cells, which prevents their spreading across other organelles and thereby preserves cellular membrane protein and lipid composition.

The Golgi apparatus is essential for the sorting of all proteins from the endoplasmic reticulum (ER) to the endo-lysosomal system, or along the secretory pathway to the plasma membrane (PM) or to the extracellular space[1–3]. Despite the pivotal role of the Golgi in protein sorting, and in cellular - and organismal health, protein quality control (PQC) processes at the Golgi have remained largely unknown. While there is emerging evidence that Golgi based PQC systems[4–7] could detect and degrade misfolded and orphaned proteins (proteins that mislocalize or fail to assemble with their protein binding partners[8,9]), the underlying molecular mechanisms are not well characterized[10–12].

A possible candidate for detecting and degrading orphaned membrane proteins at the Golgi in yeast is the 'Defective in SREBP cleavage' Dsc E3 ubiquitin ligase complex[7,13]. Yet our information on how the Dsc complex would contribute to Golgi QC is limited, since the substrate spectrum of the Dsc complex is not well defined.

The Dsc complex is a multi-subunit transmembrane protein complex, that localizes to the Golgi, endosomes, and to the limiting membrane of the vacuole. The RING E3 ligase Tul1 forms a complex with Dsc2, Dsc3, Ubx3[13,14], and with one of two trafficking adapters, Gld1 or Vld1[15]. Gld1 targets the Dsc complex to Golgi and endosomes.

[1]Institute of Molecular Biochemistry, Biocenter, Medical University of Innsbruck, Innsbruck, Austria. [2]Institute of Cell Biology, Biocenter, Medical University of Innsbruck, Innsbruck, Austria. [3]Department of General, Inorganic and Theoretical Chemistry, Center for Molecular Biosciences Innsbruck, University of Innsbruck, Innsbruck, Austria. [4]Department of Molecular and Cell Biology, University of Geneva, Geneva, Switzerland. [5]Department of Biochemistry, University of Geneva, Geneva, Switzerland. [6]Institute of Science and Technology Austria (ISTA), Am Campus 1, Klosterneuburg, Austria. [7]Institute of Medical Biochemistry, Protein Core Facility, Medical University of Innsbruck, Innsbruck, Austria. [8]These authors contributed equally: Yannick Weyer, Sinead I. Schwabl, Xuechen Tang. ✉e-mail: david.teis@i-med.ac.at

Vld1 diverts the Dsc complex from the Golgi directly to the limiting membrane of the vacuole via the AP3 pathway. Tul1 consists of an N-terminal large, luminal domain that is glycosylated, followed by seven predicted TMDs and a C-terminal cytosolic RING E3 ligase domain, which interacts with the E2 ubiquitin conjugating enzyme Ubc4[16]. Tul1 is conserved in a wide range of fungi and plants[7,17]. Direct human orthologues are not obvious[7]. Dsc2 is a rhomboid like pseudo-protease, with similarities to derlins (Der1, Dfm1) and UBA domain containing 2 (UBAC2). Dsc3 contains a UBL domain, and shares some similarities to the transmembrane and ubiquitin-like (UBL) domain containing protein 1/2 (TMUB1/2). Ubx3 has a C-terminal UBX domain that can interact with Cdc48[18], and is related to the FAS-associated factor 2 (FAF2/UBXD8). Overall, the subunit architecture of the Dsc complex resembles ubiquitin ligase complexes that function in ER-associated degradation (ERAD)[19–22].

The Dsc complex can ubiquitinate ectopically expressed, non-native proteins at Golgi and endosomes[16,23,24] as well as native vacuolar membrane proteins (mainly lysosomal transmembrane transporters)[25,26]. Once these proteins are ubiquitinated, they are recognized by the *endosomal sorting complexes required for transport* (ESCRT) and targeted into vacuoles for degradation. The Dsc complex also has the capacity to ubiquitinate phosphatidyl-ethanolamine (PE), which helps to recruit the ESCRT machinery[27].

Previously, we demonstrated that the Golgi resident Gld1-Dsc complex polyubiquitinates the ER membrane protein Orm2 on two cytosolic lysine residues (K25, K33) for Cdc48-mediated membrane extraction and proteasomal degradation. We termed this membrane protein degradation pathway *endosome and Golgi associated degradation* (EGAD)[7,20,28]. To reach the Golgi, Orm2 must no longer bind to the serine palmitoyl transferase complex (SPT, Lcb1/2) at the ER, and exit the ER in COPII carriers with the help of the derlin Dfm1[29]. Once Orm2 reaches the Golgi, it is detected by the Dsc complex, ubiquitinated, and degraded via EGAD. EGAD of Orm2 contributes to the de-repression of SPT activity and thus helps to maintain sphingolipid homeostasis[7]. Besides Orm2, the native substrate spectrum of the Gld1-Dsc complex was unknown. It was also unclear how the Dsc complex detected its substrates, and which substrates were degraded via EGAD or ESCRT pathways. Hence, the role of the Dsc complex for membrane quality control (QC) at the Golgi was not defined. Now we have closed these knowledge gaps. Here we expanded the substrate spectrum of the Dsc complex with Hmx1 and Yif1, and we show that the Dsc complex can detect short transmembrane domains (TMDs) in orphaned proteins via Dsc2 and targets them for EGAD (Orm2, Yif1, Hmx1) or ESCRT (Hmx1) mediated degradation. These results establish how the Dsc complex contributes to Golgi quality control and to cellular proteostasis in general.

## Results
### Gld1-Dsc complex substrates differ from ESCRT clients
To understand how the Dsc ubiquitin ligase complex contributed to Golgi QC, we first compared the membrane protein composition of cells lacking the E3 ligase Tul1 (*tul1Δ*) to isogenic WT cells. We used stable isotope labeling with amino acids in cell culture (SILAC) to identify changes in the membrane proteome of *tul1Δ* cells relative to WT cells (labeled with "heavy" $[^{13}C_6,^{15}N_2]$-L-lysine). We quantified 667 membrane proteins (52% of all membrane proteins), out of 3631 quantified proteins (n = 3, approximately 90% of the yeast proteome) (Fig. 1a, b and Supplementary Data 1, 2)[30]. In the *tul1Δ* cells, 21 membrane proteins were upregulated (3% of the quantified membrane proteins) (Fig. 1b, Supplementary Data 2). Many upregulated proteins were ER and Golgi proteins (Alg3, Csg2, Dgk1, Elo2, Hmx1, Orm2, Tpo5, Neo1, and Gnt1) (Fig. 1b and Supplementary Table 1). The bona fide Dsc substrate Orm2 was among the upregulated proteins[7]. A similar set of membrane proteins (Orm2, Csg2, Ftr1, Cpr8, Hmx1) was upregulated in *gld1Δ* cells, when compared to *vld1Δ* cells (n = 3, Supplementary Fig. 1a–c and Supplementary Data 1 and 2). It seemed that the Gld1-Dsc complex at the Golgi, rather than the vacuolar Vld1-Dsc complex, was required to control the levels of these proteins.

To understand how the potential Dsc substrates differed from ESCRT clients, we determined how loss of ESCRT function in *vps4Δ* mutant cells changed the membrane proteome (Fig. 1c, d and Supplementary Data 1 and 2). In total, we quantified 672 membrane proteins in the WT/*vps4Δ* cell extracts (n = 3, Fig. 1d and Supplementary Data 2). In ESCRT mutants 72 membrane proteins were upregulated (>10% of the quantified membrane proteins), of which many (34) were plasma membrane proteins (Fig. 1d, e and Supplementary Table 1), including known clients of the ESCRT machinery (e.g.: Can1, Lyp1).

Most membrane proteins that were upregulated in *vps4Δ* cells, were not upregulated in *tul1Δ* cells (and vice versa) (Fig. 1e and Supplementary Data 3). Five proteins (Hmx1, Cpr8, Csg2, Ftr1, Yml018c) were significantly upregulated in both *tul1Δ* and *vps4Δ* mutants, suggesting that these could be substrates of the Dsc complex, which are targeted to the ESCRT pathway for vacuolar degradation.

Our census of the membrane protein composition showed that the ESCRT machinery and the Dsc complex played largely complementary roles in controlling the cellular membrane protein composition.

### Dsc substrates are ER and Golgi proteins with short transmembrane domains
To identify characteristic signatures of Dsc substrates, we compared the α-helical transmembrane domains (TMDs) of the proteins that were upregulated in *tul1Δ* and *vps4Δ* mutants. TMDs were predicted using DeepTMHMM[31]. For the 21 membrane proteins upregulated in *tul1Δ* cells, DeepTMHMM predicted 125 TMDs in 18 proteins. For the 72 proteins upregulated in *vps4Δ* cells, 550 TMDs were predicted in 64 proteins (Fig. 2a, Supplementary Data 4). In *tul1Δ* cells, proteins with relatively short TMDs (hydrophobic length) accumulated: 44% of the TMDs contained 18 amino acids (Fig. 2a). 66% of the proteins had at least one TMD with 16 amino acids, and every protein had at least one TMD with 18 amino acids (Fig. 2a), with 75% of these proteins having 10 TMDs (Fig. 2a). In *vps4Δ* cells, the accumulating proteins showed a preference for longer TMDs and more TMDs per protein: Only 26% of the TMDs were short with 18 amino acids and 48% of these proteins had > 10 TMDs (Fig. 2a).

It seemed that Dsc substrates were mainly ER and Golgi membrane proteins, with relatively short TMDs, while ESCRT clients were mainly PM proteins, with longer TMDs.

### Dsc complex deficient cells upregulate short asymmetric lipids
Our findings so far are consistent with the concept that the TMDs of Dsc substrates have organelle-specific properties that match best the ER / early Golgi lipid territory with shorter and/or more unsaturated acyl chain glycerophospholipids (GPLs)[32,33]. To analyze if loss of the Dsc complex and the ensuing accumulation of ER and Golgi membrane proteins with relatively short TMDs, would cause changes in membrane lipid composition, we used shotgun mass spectrometry-based lipidomics[34]. We compared the lipid composition of WT cells to *tul1Δ* cells and *gld1Δ* cells. As an additional control we included cells in which the non-ubiquitinatable Orm2[KR] mutant accumulated (but the Dsc complex remained functional)[7]. By comparing the changes in the cellular lipid composition of these strains, we identified changes that were caused specifically by the loss of the Golgi Gld1-Dsc complex, rather than by the Orm2-mediated inhibition of SPT activity. Principal component analysis revealed that the lipid composition of *tul1Δ* cells was similar to *gld1Δ* cells, but differed from WT cells and from Orm2[KR] cells (Supplementary Fig. 2a).

The overall levels of the glycerophospholipid (GPL) classes were similar (Supplementary Fig. 2b), but *tul1Δ* and *gld1Δ* cells (but not

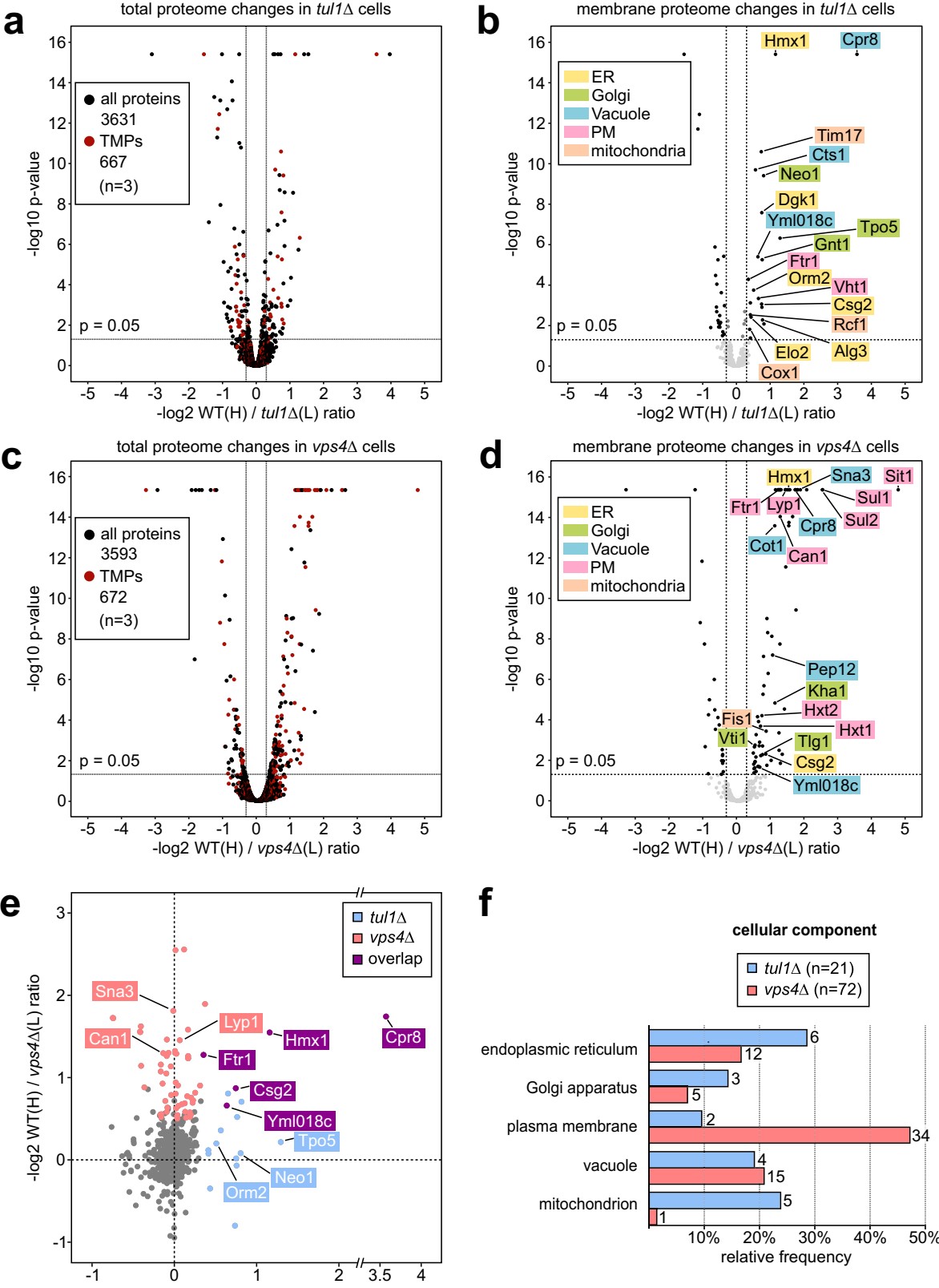

Orm2$^{KR}$ cells) showed an increase in PI, PS, PA, PC, and PE, with C-26 or C-28 fatty acyl chains (Fig. 2b, c). The levels of C-26 almost doubled and C-28 increased by 50% (Fig. 2b, c). These GPL species paired fatty acyl chains that differed substantially in length (C10; C16, or C10; C18), with either two saturated acyl chains or with the long acyl chain mono-unsaturated (Fig. 2d, e). This effect was most pronounced for PI

(Fig. 2d), and was also observed for PA, PE, and PC (Fig. 2e). Such asymmetric lipids had been reported earlier in budding yeast[34] and in fission yeast[35]. As expected in *tul1Δ* cells, in *gld1Δ* cells but also in Orm2$^{KR}$ cells, the levels of several ceramide, inositolphosphoryl-ceramide (IPC), and mannosyl-inositolphosphoryl-ceramides (MIPC) were reduced (Supplementary Fig. 2c–f)[7,28].

**Fig. 1 | Quantitative proteomic SILAC profiling of Dsc complex and ESCRT mutants. a**–**d** Volcano plots showing the H/L peptide ratios of proteins from heavy $^{13}C_6,^{15}N_2$-L-lysine labeled WT cells against light $^{12}C_6,^{14}N_2$-L-lysine L-lysine labeled *tul1Δ* **a**, **b** or *vps4Δ* **c**, **d** mutants (*n* = 3 independent experiments). Total proteome of **a** WT / *tul1Δ* mutants or **c** WT / *vps4Δ* mutants, transmembrane proteins (TMPs) in red; **b**, **d** only transmembrane proteome. Significantly regulated (−log₂ ≥ 0.3, *p* ≤ 0.05) transmembrane proteins in black. See also Supplementary Data 1, 2. **e** Scatter plot of the 614 membrane proteins quantified in WT/*tul1Δ* (x-axis) and in WT/*vps4Δ* (y-axis), only displaying proteins with −log₂ > −1. Proteins which are significantly upregulated (-log₂ ≥ 0.3, p ≤ 0.05) in both datasets are highlighted: Dsc complex-dependent (light blue), ESCRT-dependent (light red), and overlapping substrates (purple). See also Supplementary Data 3. **a**–**e** The *p*-values were calculated by using the background based *t*-test, the adjusted *p*-values by using the Benjamin-Hochberg method. **f** Gene Ontology (GO) analysis for cellular components of the significantly upregulated transmembrane proteins in *tul1Δ* (light blue) or *vps4Δ* (light red) cells. Only GO terms with ≥ 2 hits are shown as relative frequency over the data set. See also Supplementary Table 1.

Our results indicated that loss of the Golgi resident Dsc complex caused an accumulation of membrane proteins with short TMDs and an increase of short, asymmetric lipids. Hence the Golgi Dsc complex maintained cellular protein and lipid composition.

## The Dsc complex ubiquitinates the heme oxygenase 1 (Hmx1)

To understand how the Dsc complex detected and degraded its substrates, we continued our analysis with the heme oxygenase 1 (Hmx1), which was among the top 5 ranking Gld1-Dsc complex potential membrane protein substrates that were identified in our proteomic analysis (Fig. 1, Supplementary Data 2, highlighted in red). Hmx1 is a conserved ER resident membrane protein[36] that is essential for the degradation of heme. The Alphafold (AF) model[37] of Hmx1 indicated a well-defined cytosolic enzymatic fold, and a single C-terminal tail-anchored TMD (Fig. 3a).

To directly test whether Hmx1 was ubiquitinated in a Dsc complex-dependent manner, we tagged endogenous Hmx1 with an N-terminal 3xFLAG tag (hereafter FLAG-Hmx1) and immunoprecipitated FLAG-Hmx1 from cell lysates of wild-type (WT) cells, *tul1Δ*, and *dsc2Δ* mutants under denaturing conditions. Ubiquitinated FLAG-Hmx1 was recovered from WT cells, but not from *tul1Δ* and *dsc2Δ* cells (Fig. 3b). FLAG-Hmx1 migrated on an SDS−PAGE at around 38 kDa. Ubiquitinated Hmx1 was detected at about 55–130 kDa with laddering, suggesting polyubiquitination (Fig. 3b). Under non-denaturing conditions, FLAG-Hmx1 co-immunoprecipitated the four subunits of the Dsc complex Tul1, Dsc2, Dsc3, and Ubx3 (Fig. 3c). Hence, the Dsc complex interacted with Hmx1 and was essential for its polyubiquitination.

To determine how ubiquitinated FLAG-Hmx1 was degraded, we followed FLAG-Hmx1 protein turnover after blocking protein synthesis with cycloheximide (CHX). In WT cells, 70–80% of FLAG-Hmx1 was degraded after 90 min (Fig. 3d, Supplementary Fig. 3a). In cells lacking any component of the Golgi resident Dsc complex (*tul1Δ*, *dsc2Δ*, *ubx3Δ*, *dsc3Δ* or *gld1Δ*) the degradation of FLAG-Hmx1 was blocked (Fig. 3d and Supplementary Fig. 3a–c). Loss of the vacuolar Vld1-Dsc complex did not block Hmx1 turnover (Supplementary Fig. 3c).

These results showed that the Gld1-Dsc complex ubiquitinated Hmx1 for degradation.

## Ubiquitinated Hmx1 is degraded by ESCRT or EGAD pathways

We next determined if ubiquitinated Hmx1 was targeted into the EGAD pathway for proteasomal degradation or into the ESCRT pathway for vacuolar degradation. Inhibition of the proteasome with MG-132 did not impair Hmx1 degradation (Fig. 3e and Supplementary Fig. 3a). Also, the loss of ESCRT function (in *vps4Δ* cells) only slowed down Hmx1 degradation, but did not block it (Fig. 3e and Supplementary Fig. 3a). Only when proteasomal activity was inhibited in ESCRT deficient *vps4Δ* mutants, was Hmx1 degradation blocked (Figs. 3e and S3a). In these cells, we also observed the accumulation of higher molecular weight bands, possibly representing ubiquitinated Hmx1. Apparently, the Dsc complex had the capacity to target Hmx1 either into EGAD- or ESCRT-pathways for degradation.

To follow the trafficking of Hmx1 out of the ER, we replaced endogenous Hmx1 with β-estradiol-inducible mNeonGreen-ALFA-Hmx1 (NG-Hmx1)[38]. 30 or 40 min after the addition of β-estradiol, a signal for NG-Hmx1 was detected at the ER (Elo3-mCherry) (Supplementary Fig. 3d) and in some WT cells, NG-Hmx1 had already reached vacuoles (mCherry-Cps1) in WT cells (Fig. 3f and Supplementary Fig. 3d). After 90 min, NG-Hmx1 was still detected at the ER in some cells, but an increasing signal was detected in vacuoles (Supplementary Fig. 3d). After 120 min, NG-Hmx1 was mainly detected in the lumen of vacuoles in the majority of WT cells (Fig. 3f). Consistently, the protein levels of full-length NG-Hmx1 reached a plateau at 90 min, and remained relatively constant (Supplementary Fig. 3e). During the same time, the protein levels of free NG increased, which was indicative of vacuolar proteolysis (Supplementary Fig. 3e). In *tul1Δ* cells and in *gld1Δ*, NG-Hmx1 did not efficiently reach the lumen of vacuoles (Fig. 3f, g) and instead accumulated on other post-ER objects, including Sec7 positive late Golgi structures (Fig. 3f, g), and the limiting membrane of the vacuole after 90 min – 120 min of induction (Fig. 3f, g). Only a residual signal was detected in the lumen of the vacuole. Consistently, after induction, the protein levels of NG-Hmx1 were higher in *tul1Δ* cells compared to WT cells, and only little free NG was detected (Supplementary Fig. 3e). The small fraction of NG-Hmx1, that was sorted into vacuoles of *tul1Δ* cells, was likely the result of Rsp5 mediated ubiquitination of fluorescent proteins[39]. In *tul1Δ rsp5^{G747E}* (a hypomorphic allele[40]) cells, NG-Hmx1 was no longer detected in vacuoles even after 310 min induction (Supplementary Fig. 3f), while in *rsp5^{G747E}* cells, the Dsc complex efficiently targeted NG-Hmx1 into vacuoles.

It seemed that the Dsc complex ubiquitinated Hmx1 for either EGAD-mediated proteasomal or ESCRT-mediated vacuolar degradation.

## K48/K63 polyubiquitination of Hmx1 requires the Dsc complex

Since the loss of ESCRT function alone did not block Hmx1 degradation, we dissected how the EGAD pathway contributed to Hmx1 degradation in ESCRT mutants (*vps4Δ*). Therefore, we induced NG-Hmx1 expression for 90 min in *vps4Δ* cells and in *tul1Δ vps4Δ* double mutants. After the addition of CHX, the protein levels of NG-Hmx1 substantially decreased in *vps4Δ* cells, but not in *tul1Δ vps4Δ* double mutants (Fig. 4a and Supplementary Fig. 4a). Of note, free NG (released by vacuolar degradation) was not detected, suggesting EGAD mediated proteasomal degradation in *vps4Δ* cells. Consistently, 90 min after the induction, NG-Hmx1 accumulated on class E compartments in both *vps4Δ* cells and in *tul1Δ vps4Δ* double mutants, together with the ESCRT client mCherry-Cps1 (Fig. 4b). After blocking protein synthesis, the fluorescence signal for NG-Hmx1 markedly declined from class E compartments in *vps4Δ* cells and became difficult to detect, while the signal of mCherry-Cps1 persisted (Fig. 4b). In contrast, in *tul1Δ vps4Δ* double mutants the signal for NG-Hmx1 did not decrease and remained on class E compartments, with mCherry-Cps1 (Fig. 4b). It seemed that the Dsc complex mediated the degradation of NG-Hmx1 from class E compartments via EGAD[28].

An essential step in EGAD is Cdc48-mediated membrane extraction. This AAA-ATPase extracts K48 polyubiquitinated proteins from membranes. To analyze if Hmx1 was K48 polyubiquitinated, we immunoprecipitated FLAG-Hmx1 from cells under denaturing conditions, followed either by direct detection of the immunoprecipitated FLAG-Hmx1 with K48 linkage specific antibodies (Supplementary Fig. 4c) or by a second immunoprecipitation with nanobodies that are specific for K48 polyubiquitin chains

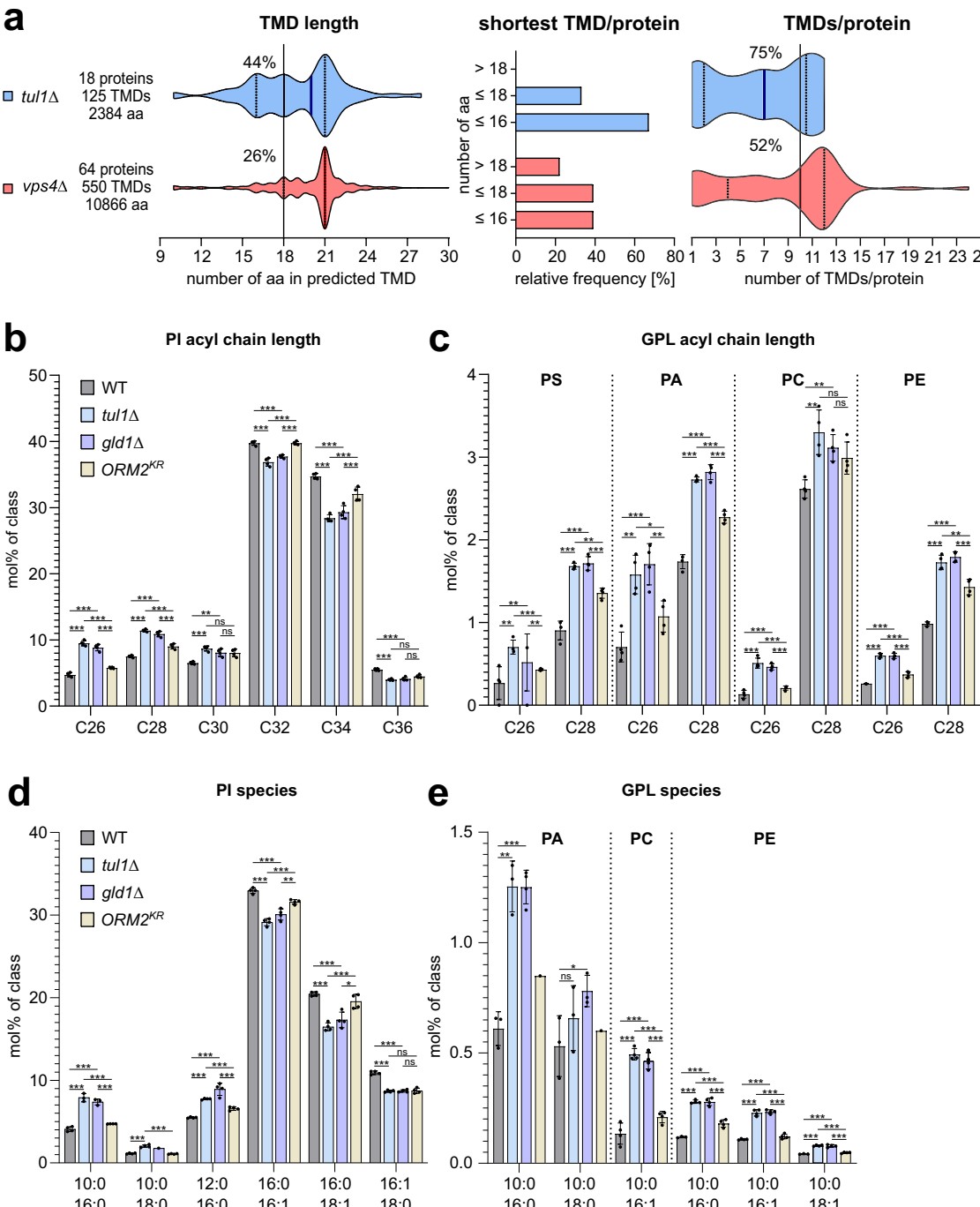

**Fig. 2 | Transmembrane domains (TMDs) of ESCRT and EGAD mutants have different signatures. a** Plots show the number of amino acids in the predicted TMDs (see Supplementary Data 4) in the upregulated proteins in *tul1Δ* and *vps4Δ*, the frequency of the shortest TMD/protein, and the number of TMDs/protein mutant cells. Bold lines indicate median value, dotted lines indicate the first and third quartile. **b**–**e** Abundances (measured using LC-MS) of **b**, **c** total acyl length species and **d**, **e** subspecies with chain composition of **b**, **d** phosphatidylinositol (PI) species, **c**, **e** phosphatidylserine (PS), phosphatidic acid (PA), phosphatidylcholine (PC), and phosphatidylethanolamine (PE) species in lipid extracts from WT (dark gray), *tul1Δ* (light blue), *gld1Δ* (soft purple), and *ORM2^{KR}* (pale yellow) mutant cells shown as mol% normalized to the respective lipid class (in mol%/class) shown as mean ± standard deviation (SD) (*n* = 4 independent experiments). Legend for all graphs is shown on the right-hand side of the figure. Data were analyzed by multiple unpaired t-test with the two-stage linear step-up procedure of Benjamini, Krieger, and Yekutieli. *P*-values > 0.05 (ns), ≤ 0.05 (*), ≤ 0.01 (**) and ≤ 0.001 (***). The exact *p*-values are provided in the source data. See also Supplementary Fig. 2.

(Supplementary Fig. 4b). The linkage specific antibodies showed that Hmx1 was K48 polyubiquitinated in WT cells (Fig. 4c), which was not picked up by the second step nanobody immunoprecipitation (Supplementary Fig. 4b). In *vps4Δ* cells, the K48 polyubiquitination of Hmx1 increased (Fig. 4c and Supplementary Fig. 4b). K48 polyubiquitination was no longer detected in the *tul1Δ vps4Δ* cells or in the *tul1Δ* cells (Fig. 4c and Supplementary Fig. 4b). Similarly,

immunoprecipitation of FLAG-Hmx1 from cells that express K48R ubiquitin as the only ubiquitin species[41], showed that the loss of K48 polyubiquitin chains almost completely abrogated Hmx1 polyubiquitination in otherwise WT cells (Supplementary Fig. 4c, lane 5) and in cells that expressed dominant negative Vps4^{E233Q} (hereafter Vps4^{EQ}) to disrupt the ESCRT pathway (Supplementary Fig. 4c, lane 6). It seemed that Hmx1 was K48 polyubiquitinated by the Dsc

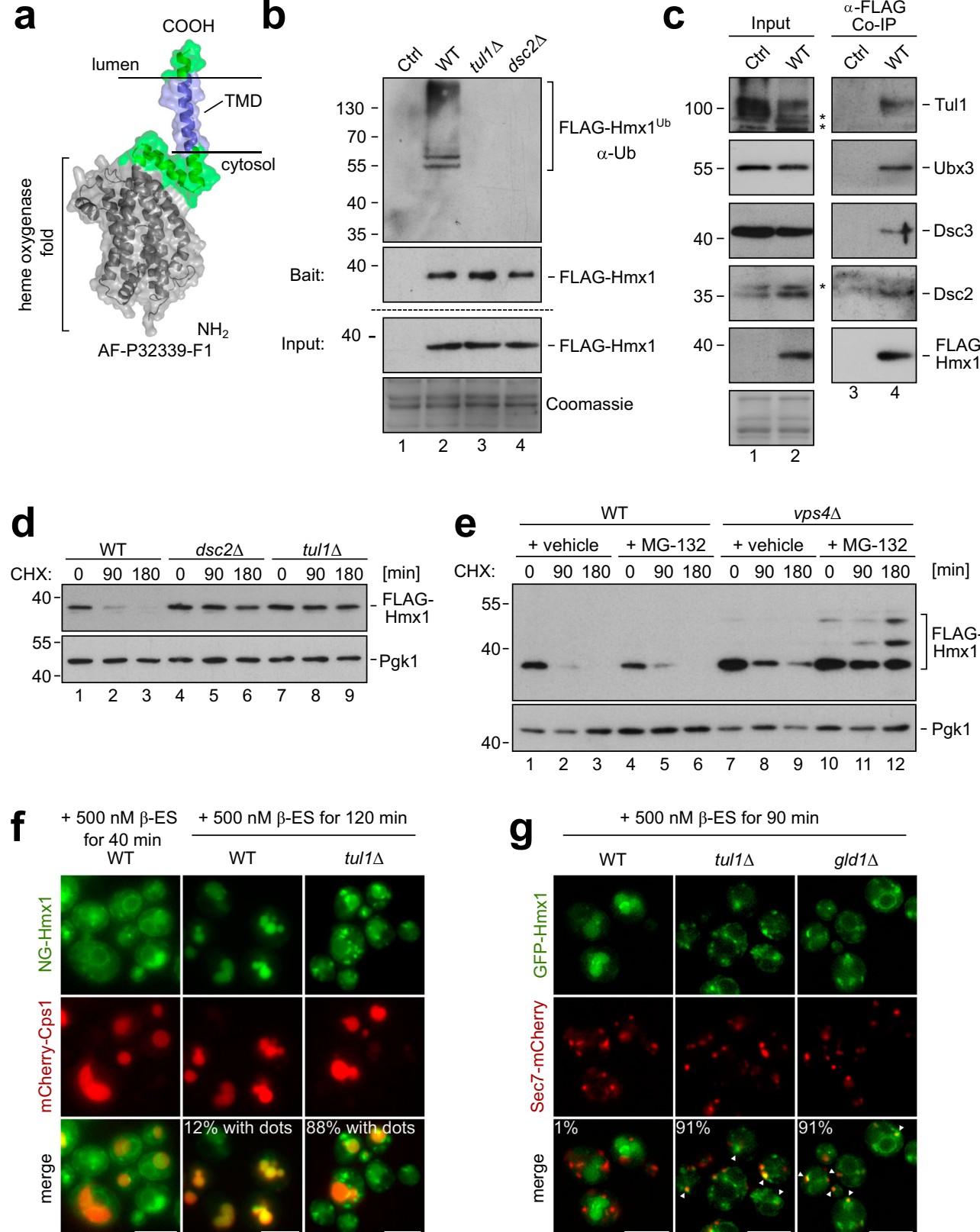

complex and that K48 polyubiquitination of Hmx1 increased in ESCRT deficient cells.

K63 polyubiquitination on immunoprecipitated Hmx1 was detected in WT using a second step immunoprecipitation with the K63 specific nanobodies (Supplementary Fig. 4b). In the *tul1Δ vps4Δ* K63 polyubiquitination was greatly reduced, but not completely

abolished (Supplementary Fig. 4b). The residual polyubiquitination showed a different laddering, and it was futile since Hmx1 was not degraded (Fig. 4a and Supplementary Fig. 4a). In cells that cannot form K63 polyubiquitin chains, Hmx1 polyubiquitination was reduced, but a significant fraction remained (probably K48) poly-ubiquitinated (Supplementary Fig. 4c). Loss of K11 polyubiquitin

**Fig. 3 | The heme oxygenase 1 (Hmx1) is ubiquitinated by the Dsc complex and degraded by ESCRT − and EGAD pathways. a** AF model of Hmx1 (AF-P32339-F1). **b**−**e** SDS−PAGE and Western blot analysis with the indicated antibodies: **b, c** Input and elution of **b** denaturing or **c** non-denaturing FLAG-Hmx1 immunoprecipitations (IP) from indicated cells. Control (Ctrl) cells were untagged WT strains. Representative blots from 3 independent experiments with similar results are shown '*' unspecific antibody cross-reactions; **d, e** Total cell lysates of the indicated cells that were untreated (0 min) or treated with 50 µg/mL cycloheximide (CHX) to block protein synthesis for the indicated times. Pgk1 served as a loading control.

Densitometric quantification is shown in Supplementary Fig. 3a. **f, g** Live cell epi-fluorescence microscopy of the indicated cells, induced with 500 nM ß-estradiol for the indicated times. **f** mNeongreen-ALFA-Hmx1 (NG-Hmx1) (green) and mCherry-Cps1 (red). Indicated are % of WT cells (n = 110) or *tul1Δ* cells (n = 119) in which dots outside of vacuoles was detected. **g** Co-localization of induced GFP-Hmx1 (green) with Sec7-mCherry (red). Indicated are % of WT cells (n = 80), *tul1Δ* cells (n = 82), *gld1Δ* cells (n = 58) in which co-localization of GFP-Hmx1 and Sec7 was detected. Scale bars 5 µm. See also Supplementary Fig. 3.

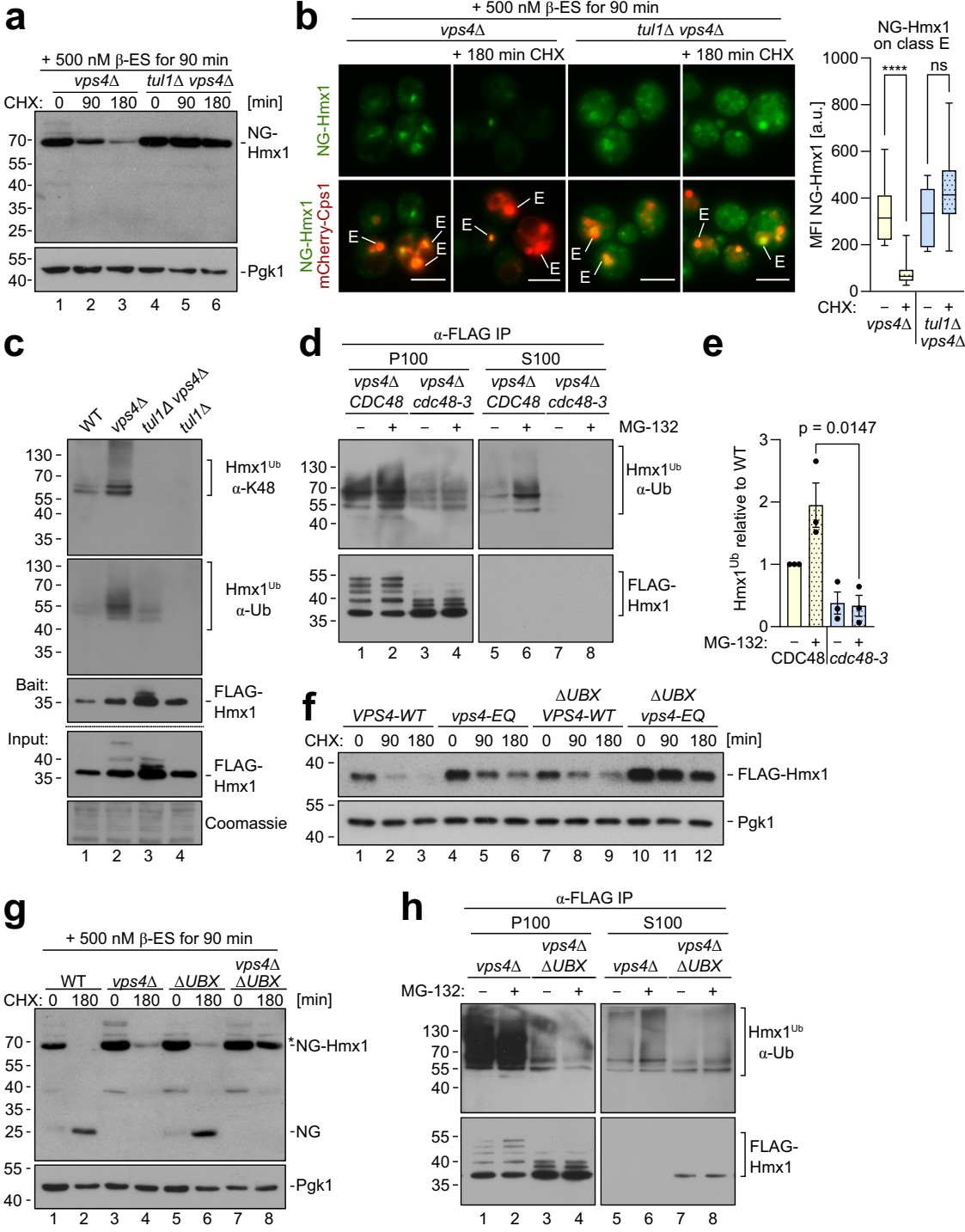

**Fig. 4 | Hmx1 is extracted from class E compartments for EGAD. a** SDS−PAGE and WB analysis with the indicated antibodies. NG-ALFA-Hmx1 (induced with 500 nM ß-estradiol (ß-ES) for 90 min) in *vps4Δ* and *tul1Δ vps4Δ* mutant cells. Cells were left untreated (0 min) or treated with CHX for the indicated time to block protein synthesis. Densitometric quantification is shown in Supplementary Fig. 4a. **b** Live cell epifluorescence microscopy: NG-ALFA-Hmx1 (green) and mCherry-Cps1 (red). Class E compartments (E) are indicated; scale bars 5 μm, **** p < 0.0001. Data were analyzed by a two-tailed unpaired t-test. Quantification of mean fluorescence intensity of NG-Hmx1 on class E compartments before and after CHX addition (*n* = 10 individual class E compartments). The mean fluorescence intensity of the individual class E compartments is presented as box plots, with lower and upper quartile and median, and with min. and max. values as whiskers. **c** SDS−PAGE and Western blot analysis with the indicated antibodies (including a K48 linkage specific antibody) of denaturing FLAG-Hmx1 immunoprecipitations (IP) from the indicated cells. **d**, **h** SDS−PAGE and Western blot analysis with the indicated antibodies of denaturing FLAG-Hmx1 immunoprecipitations (IP) from **d** *vps4Δ* cells with *CDC48* or *cdc48-3* mutants (in a *pdr5Δ vps4Δ* background) or **h** *vps4Δ* cells expressing Ubx3$^{ΔUBX}$, left untreated or treated with 50 μM MG-132. S100 and P100 fractions were subjected to denaturing FLAG-Hmx1 immunoprecipitations (IP). **e** Densitometric quantification of ubiquitinated FLAG-Hmx1 species from S100 fractions (*n* = 3 independent experiments, presented as mean ± standard error of the mean (SEM)) from Fig. 4d. Data were analyzed by a two-tailed unpaired t-test. **f**, **g** SDS−PAGE and Western blot analysis of total cell extracts from the indicated cells. **f** Cells were left untreated or treated with 50 μg/mL CHX for 90 min and 180 min. Pgk1 served as a loading control. **g** NG-Hmx1 expression was induced with 500 nM ß-estradiol (ß-ES) for 90 min prior to the addition of 50 μg/mL cycloheximide (CHX). Pgk1 served as a loading control. The densitometric quantification of (**f**, **g**) is shown in Supplementary Fig. 4d, e. For Orm2 see Supplementary Fig. 4f, g. Representative blots from 3 independent experiments with similar results are shown in **c**, **f**, **g**, **h**.

chain formation did not reduce Hmx1 polyubiquitination (Supplementary Fig. 4c).

We concluded that the Dsc complex initiated on Hmx1 the formation of K48 polyubiquitin chains for EGAD-mediated proteasomal degradation or K63 polyubiquitin chains for ESCRT-mediated degradation in vacuoles. Alternatively, mixed chains could be formed.

## Cdc48 extracts ubiquitinated Hmx1 for proteasomal degradation

Next, we tested if Cdc48 extracted ubiquitinated Hmx1 from class E compartments in *vps4Δ* cells. *vps4Δ* cells, left untreated or treated with proteasome inhibitor, were homogenized and the insoluble membrane fraction (P100) was separated from the soluble cytosolic fraction (S100) by centrifugation at 100,000×*g*[7,42,43] (Fig. 4d). FLAG-Hmx1 was immunoprecipitated from these fractions. Ubiquitinated Hmx1 predominantly accumulated in the membrane fraction (P100) (Fig. 4d, e, =3 independent experiments). As long as proteasomes were active, little ubiquitinated Hmx1 was detected in the soluble fraction (Fig. 4d, lane 5). Yet, when proteasomal degradation was blocked and Cdc48 was active, the fraction of membrane extracted polyubiquitinated Hmx1 increased in the soluble fraction (S100) (Fig. 4d, lane 6, Fig. 4e). Upon inactivation of Cdc48 (*cdc48-3*), Hmx1 was no longer detected in the soluble fraction (Fig. 4d, lane 7,8, Fig. 4e).

We considered it likely that the UBX domain of Ubx3[18] facilitated Cdc48-mediated membrane extraction. Consistent with this idea, in cells that lack the UBX domain (*ubx3$^{ΔUBX}$*), the ESCRT-mediated degradation of Hmx1 was barely impaired (Fig. 4f, lanes 7-9, Fig. 4g, lanes 3,4, and Supplementary Fig. 4d, e). Yet, when the ESCRT pathway was also disrupted in *ubx3$^{ΔUBX}$* cells, (either by expressing the dominant negative of Vps4$^{EQ}$ mutant or in *vps4Δ* cells), EGAD-mediated Hmx1 degradation was hampered (Fig. 4f, lanes 10–12, G, lanes 7,8 and Supplementary Fig. 4d, e). The appearance of a free NG band (at approximately 25 kDa) in WT and *ubx3$^{ΔUBX}$* cells suggested vacuolar turnover of NG-Hmx1 (Fig. 4g, lane 2 & 6, Supplementary Fig. 4e), and its absence indicated EGAD mediated proteasomal degradation in *vps4Δ* cells (Fig. 4g, lane 4, Supplementary Fig. 4e). In *ubx3$^{ΔUBX}$* cells, also the degradation of Orm2 was substantially delayed (Supplementary Fig. 4f) and GFP-Orm2 accumulated on post-ER compartments (Supplementary Fig. 4g). Interestingly, in *ubx3$^{ΔUBX}$* cells, the ubiquitination of Hmx1 in the membrane fraction was reduced, which could readily be observed by distinct Hmx1 laddering (Fig. 4h, lanes 3, 4), similar to Cdc48 inactivation (Fig. 4d, lanes 3, 4). In consequence, less polyubiquitinated Hmx1 was detected in the soluble fraction of *ubx3$^{ΔUBX}$* cells (Fig. 4h, lanes 7, 8), where now also non-ubiquitinated Hmx1 was detected (Fig. 4h, lanes 7, 8).

These results showed that Cdc48 extracted polyubiquitinated Hmx1 from membranes for proteasomal degradation. The UBX domain of Ubx3 was somehow involved in this process, and contributed to Hmx1 ubiquitin chain formation during membrane extraction and transfer to proteasomes.

## Dsc2 is required for substrate recognition at the Golgi

Next, we characterized how the Dsc complex interacted with its substrates. Therefore, we also included Yif1, which was reported to be a substrate of the Dsc complex when overexpressed[24]. We confirmed that endogenous Yif1 (Yif-3xHA) was a bona fide Dsc substrate, that was degraded by proteasomes via EGAD (Supplementary Fig. 5a, b).

Ubx3-FLAG co-immunoprecipitated Orm2, ALFA-Hmx1, and Yif1-HA, as well as Tul1, Dsc3, and Dsc2 from WT cells (Fig. 5a, lane 7, b). In *tul1Δ*, or *dsc3Δ* mutants, Ubx3-FLAG still co-immunoprecipitated Dsc2 and the substrates to various extents (Fig. 5a, lanes 9, 10, b). In *tul1Δ* mutants Ubx3-FLAG still retrieved Dsc3, while in *dsc3Δ* mutants, Tul1 no longer co-immunoprecipitated. Importantly, in *dsc2Δ* cells, Ubx3-FLAG failed to efficiently interact with all three substrates and with other Dsc complex subunits (Fig. 5a, lane 8, b).

Reciprocal immunoprecipitation experiments corroborated these results. In WT cells, each substrate co-immunoprecipitated all subunits of the Dsc complex (Supplementary Fig. 5c, lanes 2, 8, 14). In *dsc2Δ* cells, the interaction of the substrates with Tul1, Dsc3 or Ubx3 was strongly reduced (Supplementary Fig. 5c, lanes 4, 10, 17). In *ubx3Δ* cells, FLAG-Orm2$^{KR}$, FLAG-Hmx1, and Yif1-HA co-immunoprecipitated reduced levels of Dsc2 (Supplementary Fig. 5c, lanes 3, 9, 16). FLAG-Hmx1 and Yif1-HA still interacted with Dsc3. In *dsc3Δ* cells, all substrates still co-immunoprecipitated Ubx3 and Dsc2, but the interaction with Tul1 was reduced (Supplementary Fig. 5c, lanes 6, 12, 18). In *tul1Δ* cells, the interaction of the substrates with the remaining subunits of the Dsc complex was not affected (Supplementary Fig. 5c, lanes 5, 11, 15).

The immunoprecipitated Ubx3-Dsc2 subcomplex from *dsc3Δ* cells eluted from a size exclusion chromatography at a molecular weight that was smaller compared to that of the Dsc complex with all subunits (Fig. 5c). The molecular weight of the Dsc complex and of the Dsc2-Ubx3 subcomplex were larger than expected for a stoichiometric complex. Yet, the interpretation is difficult due to glycosylation, micelle competition, shape of the Dsc complex, or unknown additional Dsc complex subunits. Regardless, these experiments indicated that Dsc3 recruited a stable Dsc2-Ubx3 subcomplex to Tul1. It seemed that Dsc2-Ubx3 formed a substrate recognition subcomplex, as reported earlier[14,15] and that Dsc2 was essential to interact with the substrates of the Dsc complex.

Live cell fluorescence microscopy revealed that Ubx3-NG was exported from the ER in *tul1Δ* and *dsc3Δ* cells, but remained trapped at the ER in *dsc2Δ* cells[14,15] (Fig. 5d). In the vast majority of *tul1Δ* cells (93%, *n* = 75), Ubx3-NG partially co-localized with or localized to the vicinity of the early Golgi (Mnn9) (Fig. 5e). Ubx3-NG was less frequently (14%, *n* = 58) at the late Golgi (Sec7). We did not detect localization of Ubx3-NG with endosomes containing the ESCRT machinery (Vps4).

These results suggested that Dsc2 and Ubx3 were readily assembled at the ER for export. At the early/medial Golgi, the Dsc2-Ubx3

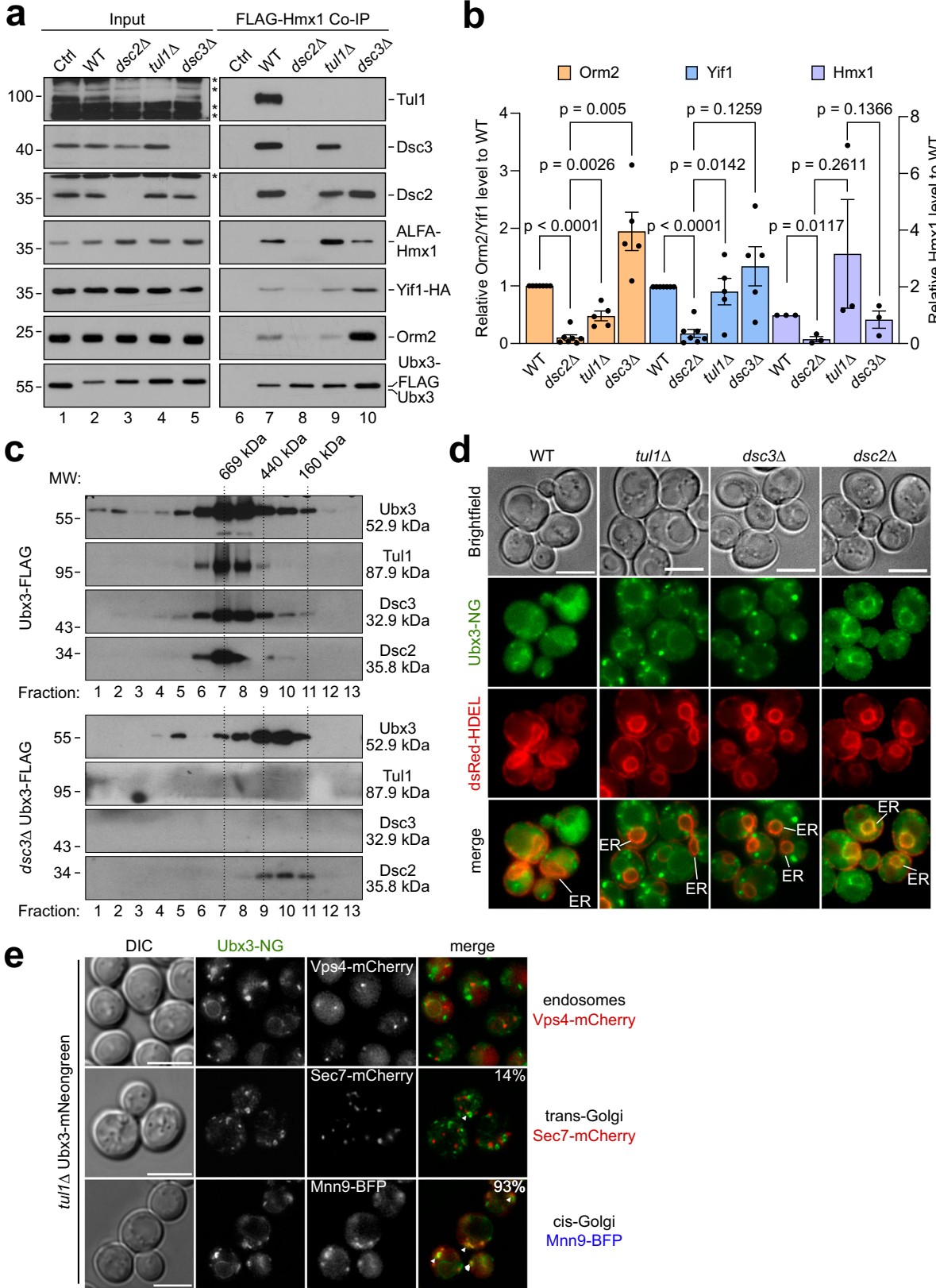

subcomplex facilitated the interaction of the Dsc complex with three different substrates.

## Dsc2 membrane thinning is required to degrade Dsc substrates

Dsc2 is a member of the rhomboid pseudoprotease family. These proteins use their rhomboid like domain to locally perturb the

membrane, which promotes membrane thinning for substrate engagement and/or retro-translocation[44–47]. To analyze if Dsc2 would also have the potential to thin membranes, we used biomolecular dynamic modeling of the rhomboid like domain of Dsc2 (AF model: AF Q08232 F1). The Dsc2 rhomboid -like domain was embedded into a lipid bilayer containing POPC (45%), POPE (45%), and cholesterol (10%)

**Fig. 5 | Dsc2 is required for substrate recognition. a, c** SDS–PAGE and Western blot analysis with the indicated antibodies: **a** elution and input from non-denaturing Ubx3-Flag immunoprecipitations from the indicated cells. Control (Ctrl) cells were untagged WT strains. Representative blots from 3 independent experiments with similar results are shown. **b** Densitometric quantification of the indicated co-immunoprecipitated substrate proteins Orm2, Yif1: WT, *dsc2Δ* (n = 7 independent experiments), *tul1Δ*, and *dsc3Δ* (n = 5 independent experiments); Hmx1 (n =n = 3 independent experiments). Data are presented as mean ± standard error of the mean (SEM). Data were analyzed by a two-tailed paired t-test. **c** Ubx3-Flag was immunoprecipitated from WT- or *dsc3Δ* cells and immunoprecipitated proteins were subjected to size exclusion chromatography. Mr (in kDa) of the proteins and the standards are indicated. **d, e** Live cell epifluorescence of WT and the indicated mutants expressing Ubx3-mNG (green) with **d** dsRED-HDEL (red). ER indicated. **e** Vps4-mCherry, Sec7-mCherry or Mnn9-BFP. In 93% of cells (n = 75) Ubx3-NG partially co-localized with or localized in the vicinity of Mnn9-BFP and in 14% of cells (n = 58) Ubx3-NG partially co-localized with or localized in the vicinity of Sec7-mCherry. Scale bars 5 μm.

and molecular dynamics (MD) were simulated for 100 ns. The MD simulation indicated that Dsc2 had the capacity to thin membranes between the L1 loop and TMD2 and TMD5 (Fig. 6a, Supplementary Fig. 6a, Supplementary movies 1, 2). Calculating the average center of mass between the membrane leaflets during the simulation and displaying it as a lipid density map revealed that, in this region of Dsc2, the lipid bilayer was thinned by about 5 Å (Fig. 6b, Supplementary Fig. 6a, Supplementary movie 3). Since asymmetric lipids were upregulated in *tul1Δ* − and *gld1Δ* cells, we used MD simulation to test if asymmetric lipids interacted with Dsc2. In simulations, Dsc2 gradually approached the saturated C10:0 intermediate length acyl chain of PC (10:0; 16:0) (Supplementary Fig. 6b, c) and stably interacted with it for the rest of the simulation (40 ns) (Supplementary Fig. 6c), but not with a mono-unsaturated C10:1 acyl chain or POPC (18:1, 16:0).

To analyze how Dsc2 membrane thinning activity contributed to substrate degradation, we introduced point mutations in the conserved L1 loop region of Dsc2 (Y53A, R54A)[46,47] (Supplementary Fig. 6d). In MD simulations, the Dsc2$^{Y53A,R54A}$ mutant reduced the local membrane thinning capacity of the mutant L1 loop by 2,5 Å (Fig. 6a, b and Supplementary Fig. 6a), which disrupted Dsc2 function and no longer supported the degradation of Orm2 (Fig. 6c and Supplementary Fig. 6e) and Hmx1 (Fig. 6d and Supplementary Fig. 6f). Consequentially, GFP-Orm2 accumulated in post-ER puncta in the Dsc2$^{Y53A,R54A}$ mutant (Fig. 6e). Likewise, after 120 min of induction, NG-Hmx1 was no longer sorted into the vacuole in Dsc2$^{Y53A,R54A}$ mutants and also started to accumulate in punctate structures (Fig. 6f).

These results demonstrated that the membrane thinning capacity of Dsc2 was required for the degradation of Dsc complex substrates.

## The TMD of Hmx1 is a degron and interacts with Dsc2

Since Dsc2 required its capacity to thin membranes for substrate degradation and because TMD length was a discriminator for Dsc complex substrates, we predicted that TMD length defined Dsc complex degrons. To test this prediction, we generated a model substrate using the single C-terminal TMD (18 aa, from aa residue 293–310) of Hmx1, including a short cytosolic α-helix with three lysine residues (K265, K269, K282), encompassing amino acid residues 264–317 (hereafter Hmx1$^{264}$) (Fig. 7a). We generated two additional model substrates by extending N-terminally the hydrophobic length of the TMD by 3 (LIV) or 6 amino acids (LIVLIV), which added an additional pitch of around 4,5 Å and 9 Å, respectively. Hmx1$^{264}$ (WT), Hmx1$^{264+3}$ (N + 3) or Hmx1$^{264+6}$ (N + 6) (Fig. 7a), were N-terminally fused to mNeonGreen-ALFA-(NG)-tags or GFP-ALFA-(GFP)-tags. When the model substrates where constitutively expressed, the protein levels of the model substrates with the longer TMDs (N + 3, N + 6) were higher compared to GFP-Hmx1$^{264}$ (Supplementary Fig. 7a). The clipped GFP suggested that these proteins were still sorted into vacuoles. To directly compare their turnover, we induced the expression of the model substrates and followed their degradation. After induction for 90 min, the protein levels of the three different model substrates were similar (Fig. 7b and Supplementary Fig. 7b). Upon addition of cycloheximide, NG-Hmx1$^{264}$ protein levels rapidly declined within 60–120 min (Fig. 7b, lanes 1–3 and Supplementary Fig. 7b), while the degradation of the longer model substrates N + 3 and N + 6 was markedly delayed (Fig. 7b, lanes 4-9 and Supplementary Fig. 7b). To

compare the interaction of these model substrates with the Dsc complex, we immunoprecipitated GPF-Hmx1$^{264}$, N + 3, and N + 6. This experiment was conducted from *tul1Δ* cells, which ensured that the steady state protein levels of the different model substrates were comparable. Notably, the longer model substrates interacted less efficiently with Dsc2, Dsc3, and Ubx3 (Fig. 7c and Supplementary Fig. 7c).

To pinpoint if the TMD of Hmx1 contained a Dsc complex-specific degron, we determined the turnover of GPF-Hmx1$^{264}$ in WT cells and in *tul1Δ* cells. In WT cells the protein levels of GFP-Hmx1$^{264}$ were low and declined within 60 min (Supplementary Fig. 7d). In *tul1Δ* cells, GFP-Hmx1$^{264}$ levels were readily elevated and the protein was not degraded (Supplementary Fig. 7d). Consistently, live cell fluorescence microscopy showed, that GFP-Hmx1$^{264}$ (Fig. 7d) or pHluorin-Hmx1$^{264}$ (Supplementary Fig. 7e) accumulated in dot-like objects, probably Golgi and endosomes, and on the limiting membrane of vacuoles in *tul1Δ* cells (Fig. 7d and S7e). In WT cells, GFP-Hmx1$^{264}$ localized mainly to the lumen of FM4-64 labeled vacuoles (Fig. 7d) and pHluorin-Hmx1$^{264}$ was quenched inside vacuoles but detected at structures reminiscent of nuclear and cortical ER (Supplementary Fig. 7e).

The cytosolic lysine residues of GFP-Hmx1$^{264}$ were polyubiquitinated in a Dsc complex-dependent manner. When the three cytosolic lysine residues were mutated to arginine (K265R, K269R, K282R), GFP-Hmx$^{264-KR}$ accumulated, but polyubiquitination was no longer detected (Fig. 7e), similar to *tul1Δ* cells. Only a small fraction of apparently mono- or oligo-ubiquitinated GFP-Hmx1$^{264}$ was detected (Fig. 7e). Thus, K265, K269, and K282 are the major targets for Dsc complex mediated polyubiquitination and degradation.

These lysine residues were K48 polyubiquitinated in WT cells and in *vps4Δ* cells by the Dsc complex (Supplementary Fig. 7f, compare lanes 1 and 2 with lane 3). The Dsc complex (Tul1) was also essential for the degradation of GFP-Hmx1$^{264}$ from class E compartments in ESCRT mutants, presumably via EGAD (Supplementary Fig. 7g, h). Consistent with EGAD-mediated degradation, GFP-Hmx1$^{264}$ showed little or no co-localization with FM4-64 on class E compartments, when the ESCRT machinery was disrupted by expression of a dominant negative Vps4$^{EQ}$ (Supplementary Fig. 7h). Yet, when the UBX domain of Ubx3 was deleted, to interfere specifically with EGAD, GFP-Hmx1$^{264}$ was no longer efficiently extracted from class E compartments, and now accumulated there with FM4-64 (Supplementary Fig. 7h).

The model substrate GFP-Hmx1$^{284}$ (aa residue 284–317), which encompassed the TMD of Hmx1, but lacked the cytosolic α-helix with the three ubiquitinated lysine residues, was no longer degraded (Supplementary Fig. 7d, lanes 7–9). Yet, GFP-Hmx1$^{284}$ still co-immunoprecipitated all subunits of the Dsc complex (Fig. 7f, lane 2) and this interaction required Dsc2 (Fig. 7f, lanes 4). To analyze if Dsc2 would be sufficient for the interaction with GFP-Hmx1$^{284}$, we replaced the endogenous promotor of *DSC2* with a galactose inducible promotor for strong over-expression in *dsc3Δ* cells. Indeed, over-expressed Dsc2 co-purified with GFP-Hmx1$^{284}$ (Fig. 7g), demonstrating that excess Dsc2 alone has the capacity to interact with the TMD degron of Hmx1.

Finally, we asked if the TMD degron of Hmx1 was transposable. Towards this goal we generated a chimeric protein, consisting of the

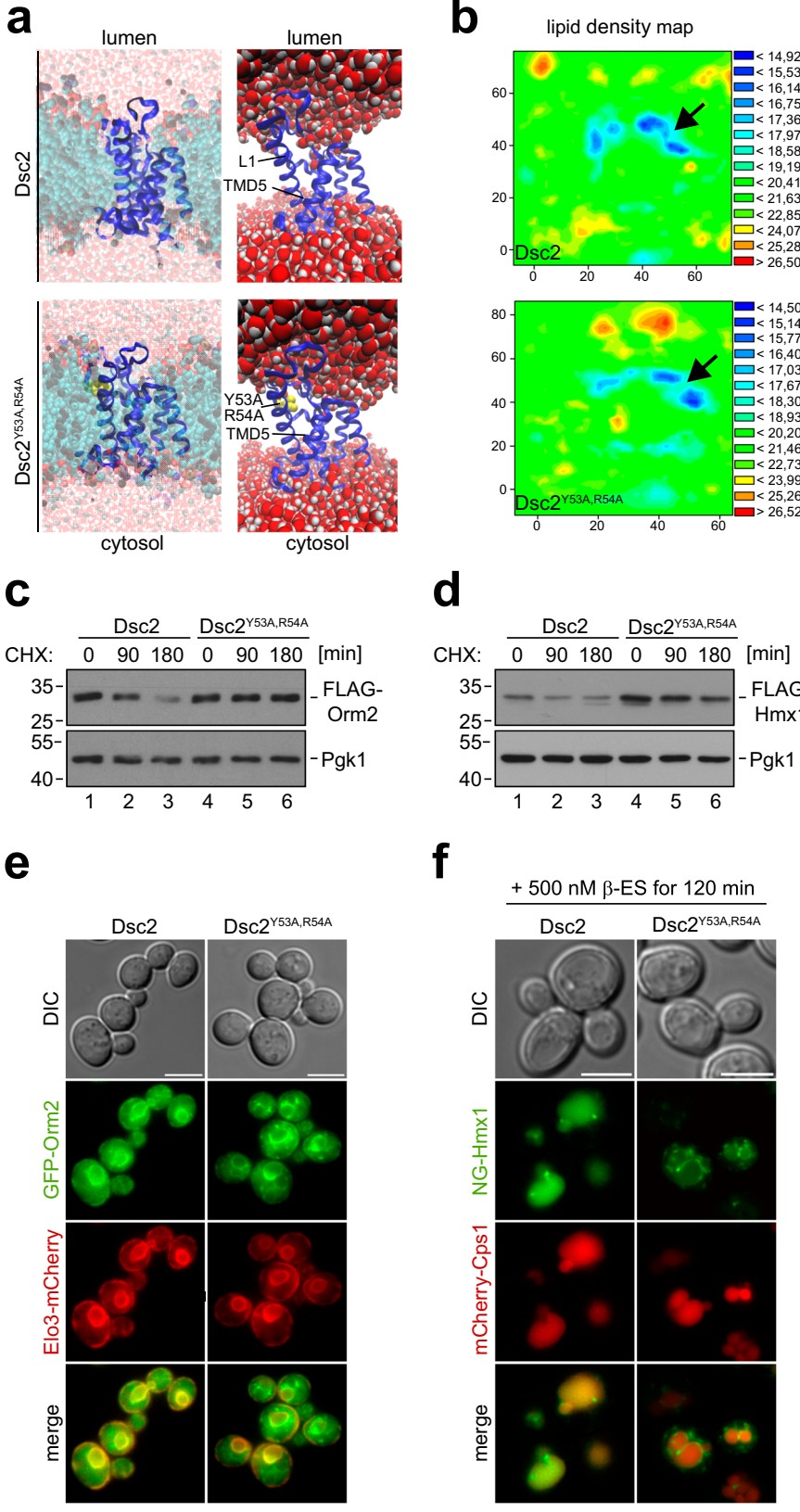

cytosolic portion of Pep12 (an endosomal t-SNARE) and the TMD (residues 284–317) of Hmx1. Only the GFP-Pep12-Hmx1[284] chimera, but not the GFP-Pep12[16], was degraded in a Dsc complex-dependent manner (Fig. 7h, Supplementary Fig. 7i).

Our results demonstrate that the TMD of Hmx1 is a degron for the Dsc complex and that Dsc2 is required to interact with this degron,

probably by assessing its hydrophobic length. Based on the proteomic dataset and simulations, short TMDs are likely regulated by the Dsc complex as degrons in general. Thereby, the Dsc complex detects orphaned membrane proteins (Fig. 7i), prevents their accumulation at the Golgi, and restricts their spreading from the Golgi to other organelles.

**Fig. 6 | The membrane thinning capacity of Dsc2 is required for substrate degradation. a** Snapshot of the molecular dynamics simulation of the rhomboid domain of Dsc2 (see also Supplementary movies 1 and 2) or Dsc2$^{Y53A,R54A}$ showing the entire system (including lipids and water) or just the Dsc2 rhomboid domain with water in orthographic and perspective views. **b** Lipid density maps of Dsc2 (see also Supplementary movie 3) or Dsc2$^{Y53A,R54A}$ displaying the average distances (in Å) of the center of mass in the lipid bilayer during the final converged 40 ns of the simulation. **c**, **d** SDS−PAGE and WB analysis with the indicated antibodies for the indicated cells, left untreated (0 min) or treated with CHX for the indicated time to block protein synthesis. Representative blots from three independent experiments with similar results are shown. Densitometric quantification is shown in Supplementary Fig. 6e, f. **e**, **f** Live cell epifluorescence microscopy: **e** GFP-Orm2 (green) and Elo3-mCherry (red); **f** ß-ES induced NG-ALFA-Hmx1 (green) and mCherry-Cps1 (red). Scale bars 5 μm. Representative micrographs from three independent experiments with similar results are shown.

## Discussion

Our work shows that the Dsc ubiquitin ligase complex plays a central role in membrane quality control at the Golgi, where it detects orphaned proteins and ubiquitinates them for degradation (Fig. 7i). Our model includes the following steps: (1) Dsc2-Ubx3 assemble at the ER and are required for substrate recognition at the Golgi. (2) Substrates are typically ER and Golgi membrane proteins, and their shorter TMDs could be detected by the Dsc complex. (3) Dsc2 could interact preferentially with such shorter TMDs and uses its L1 loop to locally thin the membrane for substrate degradation. (4) The UBX domain of Ubx3 critically contributes to Cdc48-mediated membrane extraction of substrates and to their transfer to proteasomes. (5) Dsc3 connects Ubx3-Dsc2 with the E3 ubiquitin ligase Tul1. (6) Tul1 initiates the conjugation of polyubiquitin chains onto the substrates.

The ubiquitination of all known Dsc substrates (including PE) requires the canonical enzymes Uba1 (E1), Ubc4/5 (E2), and Tul1 (E3)[13,16,24,27]. Therefore, we propose that Tul1-Ubc4 sets a priming mono- or oligo-ubiquitination, that can be extended into different multi- or polyubiquitin chains. For EGAD, the conjugated ubiquitin residues are elongated into K48 polyubiquitin chains. Perhaps this step is facilitated by the UBX domain of Ubx3, which might coordinate Cdc48 to bring along Ufd2 as an E4 ubiquitin ligase to extend the priming ubiquitination mediated by Tul1-Ubc4 into a K48 polyubiquitin chain. This would reinforce Cdc48-mediated membrane extraction, and/or transfer to proteasomes for degradation or both[48–50]. Orm2 and endogenous Yif1 follow this pathway.

For ESCRT-mediated vacuolar degradation, ubiquitinated residues are either not elongated or elongated into K63 polyubiquitin chains. Elongation could involve other E3 ubiquitin ligases such as Rsp5 or Pib1[51,52].

The formation of the different polyubiquitin chains might depend on how the Dsc complex interacts with its substrates. Substrates with shorter TMDs may be better hydrophobically matched with Dsc2, which could increase the dwell time and, thus, successful Cdc48 engagement and membrane extraction, while faster dissociation from Dsc2 could result in the sorting of the ubiquitinated substrates into the ESCRT pathway. Perhaps conceptually related is the phenomenon that substrates that escape ERAD, are degraded in an ESCRT-dependent fashion[53,54]. It is currently unknown if the Dsc complex contributes to the detection of ERAD escapers and targets such proteins for degradation.

The detection or the extraction of TMDs might be facilitated by the capacity of Dsc2 to locally perturb the lipid bilayer[44,45], similar to the function of derlins in ERAD[46,47]. While our results show that Dsc2-mediated membrane thinning is required for substrate degradation, it might do so either by promoting hydrophobic matching with the potential TMD degrons and/or additionally destabilize TMDs to overcome the energetic barrier during retro-translocation, for efficient ubiquitination and Cdc48-mediated membrane extraction[55–57].

For Hmx1, the tail-anchored TMD functions as a degron that is detected by Dsc2 at the Golgi.

For Orm2, we propose that the potential degrons could be TMD3 (14 amino acids) and/or TMD4 (18 amino acids). These rather short TMDs would become exposed, as soon as Orm2 no longer interacts with Lcb1 of the SPT complex[58,59]. Similarly, Yif1 appears to interact via its short TMD1 (18 amino acids) with Yip1 (based on AF models[37]). Hence, for orphaned Yif1, the TMD1 might contain a degron that could be detected by Dsc2. Alternatively, exposure of the degron might occur when Yif1 is idle or when orphaned Yif1 departs from its regular ER-cis-Golgi route into the trans-Golgi.

The spatial separation of the Dsc complex to the Golgi would prevent the premature degradation of free Orm2 and Yif1 at the ER. This could improve the efficiency of Orm2-SPT or Yif1-Yip1 complex assembly, while at the Golgi it reduces the level of orphaned Orm2 and Yif1. A similar concept was proposed for the quality control of unassembled membrane proteins by Asi-ERAD at the inner nuclear membrane[60].

It remains to be tested if the detection of human orthologues of Orm2 (ORMDL1-3)[61], Yif1 (YIPF3), and Hmx1 (HO-1)[62,63] is conceptually similar. However, it is already clear that ORMDL and HO-1 are ubiquitinated, membrane extracted, and degraded by proteasomes. The degradation of HO-1 requires MARCH6/TEB4 and TRC8 in a redundant manner together with the signal peptide peptidase (SSP)[62]. The same study also found ORMDL2 among the upregulated proteins in MARCH6/TRC8 double knockout cells.

In Dsc complex deficient cells, Orm2, Yif1, and Hmx1 accumulate at the Golgi, on endosomes, and the limiting membrane of vacuoles. Interestingly, they were excluded from the plasma membrane (PM), presumably due to hydrophobic mismatches with the thicker lipid bilayer of the PM[33,64,65]. Hence, our results are in agreement with previous work that showed that physical properties of TMDs, such as length, hydrophobicity, and volume, function as sorting signals within the lipid bilayers of eukaryotic cells and must be matched by the hydrophobic core of the lipid bilayer[32,35,66–69]. Our work now suggests that in addition to being a sorting signal, similar physicochemical properties of TMD properties can function as degrons to flag orphaned proteins. Whether and how the increase in shorter, and asymmetric GPL[34,35] is part of a membrane stress response[70] that could help to accommodate the accumulating Dsc substrates by improving hydrophobic matching is an important question for future studies.

Overall, the Dsc complex appears to be optimally positioned for membrane quality control at the Golgi apparatus, where it plays a crucial role in integrating lipid bilayer thickness with biochemical/biophysical properties of TMDs, such as hydrophobic length. Such hydrophobic matching of TMDs may serve as a general mechanism for protein quality control systems to identify and degrade orphaned proteins, thereby preventing their spread throughout the cell.

## Methods

### Yeast strains, plasmids, reagents, and growth conditions

*S. cerevisiae* strains in this study were SEY6210 derivatives, except for the ubiquitin linkage deficient strains and the galactose inducible *DSC2* strain. All yeast strains, plasmids and reagents are listed in Supplementary Table 2. For liquid cultures, cells were incubated in YNB synthetic medium supplemented with amino acids and 2% glucose at 26 °C in a shaker and were grown to midlog phase (OD$_{600}$ = 0.5 − 1) for all experiments. Temperature sensitive mutants were shifted to restrictive temperature (37 °C or 30 °C). Expression of galactose inducible *DSC2* was done by culturing the cells for 16 h in YNB synthetic

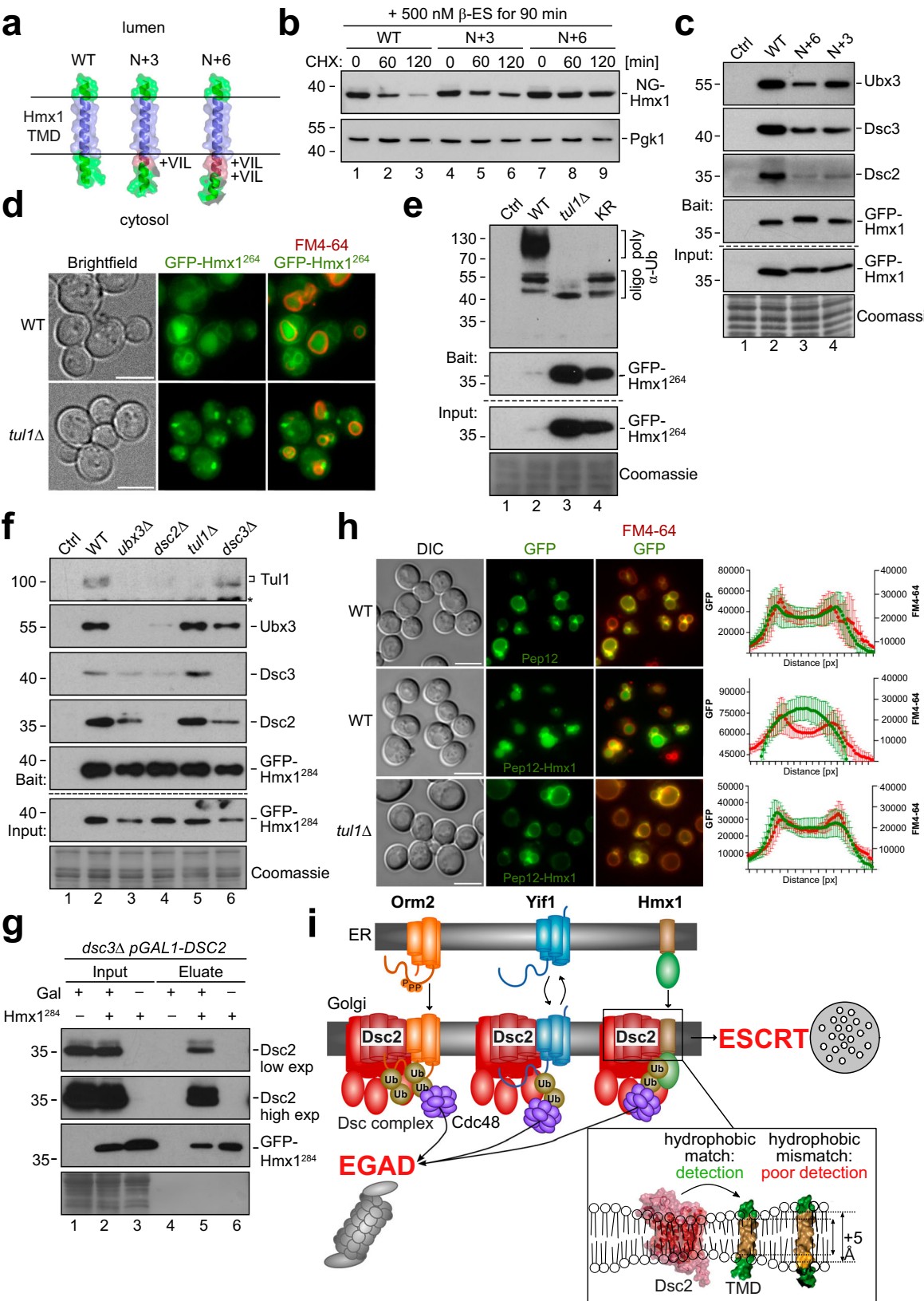

medium supplemented with amino acids and 2% galactose. Ubiquitin deficient strains were cultured in YP + 2% galactose supplemented with 200 ng/mL G418 and 100 ng/mL Nourseothricin. To activate the expression of mutant ubiquitin forms, cells were switched to YP + 2% glucose supplemented with 100 μM CuSO₄, 200 ng/mL G418, and 100 ng/mL Nourseothricin for 6 h prior to harvest.

Genetic modifications were performed by PCR and/or homologous recombination using standard techniques. Plasmid-expressed genes including their native promoters and terminators were amplified from yeast genomic DNA and cloned into centromeric vectors (pRS series). All constructs were analyzed by DNA-sequencing and transformed into yeast cells using standard techniques. ß-Estradiol

**Fig. 7 | The TMD of Hmx1 is a degron for the Dsc complex. a** AF models (AF-P32339-F1) showing the region including the TMD of Hmx1 from amino acid (aa) 284–317 (WT), 284 + 3 aa (N + 3), and 284 + 6 aa (N + 6). Transmembrane domain (TMD) (blue) aa 293–310 and extended α-helices (red) are colored as indicated. These constructs also encompass aa 264–317, but the cytosolic helix with aa 264–284 (including K265,269,282) is not shown. **b**, **c**, **e**–**g** SDS–PAGE and Western blot analysis with the indicated antibodies of the indicated mutants expressing the model substrates. Coomassie and Pgk1 served as a loading control. '*' indicates cross-reactive bands. **b** NG-Hmx1 was induced for 90 min and cells were left untreated (0 min) or treated with 50 μg/mL cycloheximide (CHX) to block protein synthesis for the indicated time points. Densitometric quantification **in** Supplementary Fig. 7b. **c** Immunoprecipitations of different model substrates from *tul1Δ* cells. Control (Ctrl) cells were untagged WT cells. Densitometric quantification **in** Supplementary Fig. 7c. **d**, **h** Live cell epifluorescence of indicated cells expressing the **d** model substrate GFP-Hmx1²⁶⁴ (green) or **h** GFP-Pep12 (green, upper panel) or GFP-Pep12-Hmx1 (green, middle and lower panel) with FM4-64 (red, vacuolar membrane). Quantification of fluorescence intensity across 10 individual vacuoles is shown and presented as mean ± standard deviation (SD). Scale bars 5 μm. **e**–**g** Input and elution of **e** denaturing GFP-Hmx1²⁶⁴ immunoprecipitations (IP) or **f**, **g** non-denaturing GFP-Hmx1²⁸⁴ from the indicated cells. Control (Ctrl) cells were untagged WT strains or **g** uninduced cells. Representative blots from three independent experiments with similar results are shown in **b**, **c**, **e**–**g**. **i** Summary of Dsc complex mediated degradation of orphaned transmembrane proteins. Structural models of Dsc2 (AF Q08232 F1) and the TMD of Hmx1 (AF P32339 F1) were generated using AF.

inducible plasmids are based on RB3579 and are derivatives of non-inducible pRS series plasmids. In detail, the entire ORF of yeGFP was excised with XhoI and KpnI, and replaced with the desired ORF of the gene of interest, which was flanked by XhoI and KpnI restriction sites. The desired ORF was amplified from existing plasmids with a forward primer that contained the starting methionine and the downstream 3' region of M1, and a reverse primer that annealed within the 3'UTR of the respective gene of interest. Endogenously FLAG-tagged copies of *ORM2/HMX1* were cloned into pFA6a-*TRP1* plasmids and amplified by PCR with primers homologous to the respective promoter and 3'UTR of the gene of interest. This generated a cassette containing the gene of interest with the *TRP1* selection marker. This product was then integrated by homologous recombination into SEY6210.1 strains containing either a disrupted *orm2::URA3* or *hmx1::HIS3* locus. Integrants were counter selected and verified with DNA-sequencing. All yeast strains and plasmids used in this study as well as primer for PCR-based genetic modifications and cloning are listed Supplementary Table 2. pRS416-yZ3EV-Z3pr-yEGFP (RB3579) was a gift from David Botstein (Addgene plasmid # 69100; RRID:Addgene_69100). YIplac211-SEC7-mCherry2Bx6 was a gift from Benjamin Glick (Addgene plasmid # 105267; RRID:Addgene_105267). ZJOM100 was a gift from Zhiping Xie (Addgene plasmid # 133662; RRID:Addgene_133662).

### SILAC labeling and sample preparation for mass spectrometry

For quantitative analysis of protein abundances *tul1Δ*, *vps4Δ* and *gld1Δ* mutants and their respective plasmid rescued WT cells (or *vld1Δ* in the case of *gld1Δ*) were grown in synthetic medium (YNB) containing 2% glucose with all required amino acids except for uracil and lysine. *vld1Δ* and *gld1Δ* cells were grown in the presence of uracil. Cells were pre-cultured for 24 h in 10 mL medium at 26 °C. WT and *vld1Δ* cells were grown with "heavy" [$^{13}C_6^{15}N_2$]-L-lysine (Sigma) at a final concentration of 20 mg/L and the *tul1Δ*, *vps4Δ* and *gld1Δ* mutant cells were grown with "light" [$^{12}C_6^{14}N_2$]-L-lysine at a final concentration of 20 mg/L. Cells were grown to mid-log phase (OD600 ≤ 0.8) for 12 generations (24 h). After two washes at 4 °C with cold YNB medium without amino acids, WT "heavy" cells were mixed with an equal number of "light" mutant cells, filtered over a nitrocellulose membrane with a pore size of 0.2 μm (Amersham) and frozen in liquid nitrogen. Frozen cells were ground in a 6770 freezer mill (SPEX Sample Prep) with liquid nitrogen cooling and ground material was stored at −80 °C. For each experiment 3 different biological replicates ($n = 3$) were analyzed.

An aliquot of 1 mg dried cell lysate was dissolved in 200 μL ammonium hydrogen carbonate buffer (100 mM, pH 8) containing 0.5 % (w/w) sodium deoxycholate (SDO). The proteins were reduced by adding 20 μL of 100 mM DTT (30 min at 56 °C), alkylated with 20 μL 550 mM IAA (20 min at RT), and quenched with 20 μL 1 M DTT. Half of the sample was digested with 10 μg trypsin (Promega, 18 h at 37 °C). DOC was precipitated with 10 μL concentrated formic acid (conc. FA) and centrifuged at 36,000 × g for 10 min (supernatant 1). The pellet was redissolved in 40 μL 1 M ammonium solution,

reprecipitated with 7 μL conc. FA and centrifuged at 36,000 × g for 10 min (supernatant 2). The two supernatants were combined and subjected to high pH fractionation using Pierce High pH Reversed-Phase Peptide Fractionation Kit (Thermo Scientific), according to the manufacturer's instructions.

### LC-MS/MS analysis

High pH fractions were analyzed using an UltiMate 3000 nano-HPLC system coupled to a Orbitrap Eclipse mass spectrometer (Thermo Scientific, Bremen, Germany). The peptides were separated on a homemade fritless fused-silica microcapillary column (100 μm i.d. × 280 μm o.d. × 16 cm length) packed with 2,4 μm reversed-phase C18 material (Reprosil). Solvents for HPLC were 0.1% formic acid (solvent A) and 0.1% formic acid in 85% acetonitrile (solvent B). The gradient profile was as follows: 0–4 min, 4% B; 4–117 min, 4–30% B; 117–122 min, 30-100% B, and 122–127 min, 100 % B. The flow rate was 300 nL/min.

The Orbitrap Eclipse mass spectrometer equipped with a field asymmetric ion mobility spectrometer (FAIMS) interface was operating in the data dependent mode with compensation voltages (CV) of −45, −55 and −75 and a cycle time of one second. Survey full scan MS spectra were acquired from 375 to 1500 m/z at a resolution of 120,000 with an isolation window of 1.2 mass-to-charge ratio (m/z), a maximum injection time (IT) of 50 ms, and automatic gain control (AGC) target 400 000. The MS2 spectra were measured in the Orbitrap analyzer at a resolution of 15,000 with a maximum IT of 22 ms, and AGC target or 150 000. The selected isotope patterns were fragmented by higher-energy collisional dissociation with normalized collision energy of 30%.

### Analysis of quantitative proteome data and TMD analysis

Data Analysis was performed using Proteome Discoverer 2.5 (Thermo Scientific) with search engine Sequest. The raw files were searched against yeast database (orf_trans_all.fasta) with trypsine as protease which cleaves proteins C-terminal at lysine and arginine, except for Arg-Pro and Lys-Pro bonds. Precursor and fragment mass tolerance was set to 10 ppm and 0.02 Da, respectively, minimum peptide length was set to six amino acids, and up to three missed cleavages were allowed. Carbamidomethylation of cysteine was set as static modification, oxidation of methionine and heavy lysine was set as variable modifications. Protein N-terminal variable modifications were acetylation, methionine-loss, and acetylation after methionine loss. Peptide identifications were filtered at 1% false discovery rate. Three independent biological replicates were measured and analyzed. The source data of all proteomics experiments are available via ProteomeXchange with identifier PXD055553.

For further analysis only proteins with two or more H/L peptides that were quantified in at least two of the three biological experiments were considered. Proteins with a -log₂ H/L ratio ≥ 0.3 and a *p*-value ≥ 0.05 were considered as significantly upregulated and compared against the landscape of yeast membrane proteins[30]. Proteins which fulfilled these criteria were submitted to the DeepTMHMM server[31],

subsequently the amino acid sequence of the transmembrane domains of membrane proteins were isolated and aligned with respect to their bilayer orientation, from the cytosolic side to the luminal side, with a custom-made R script.

Gene ontology (GO) term analysis was performed using the upregulated transmembrane standard names. They were mapped against the generic component terms using the GoSLIM mapper of the *Saccharomyces* genome database. GO terms with only one hit were removed and hits from the term 'cytoplasmic vesicles' were fused either with the plasma membrane or with endosomes according to their localization in the yeast GFP database. The ratio of the observed (dataset frequency) vs. the expected number of genes (genome frequency) associated with the GO term was also calculated and referred to as enrichment over genome.

## Molecular dynamics simulation
Details and references for the MD simulation are described in detail in Supplementary Methods and in Supplementary Table 3.

## Preparation of whole cell protein extracts
To prepare whole cell lysates, proteins were extracted by a modified alkaline extraction protocol[7]. Briefly, 4 $OD_{600}$ of logarithmically growing yeast were harvested by centrifugation (5 min, 3,200 x g, 4 °C) and washed in ice cold 10 mM NaF. After centrifugation (3 min, 21,000×*g*, 4 °C) cells were resuspended in 0.1 M NaOH and incubated at room temperature for 5 min. After centrifugation (3 min, 21,000 x g, 4 °C) pellets were resuspended in Lämmli sample buffer (60 mM Tris-HCl pH 7.5, 2% SDS, 1% β-mercaptoethanol, 10% glycerol, bromophenol blue), denatured (95 °C, 5 min), and cell debris was removed by centrifugation (3 min, 21,000 x g, 4 °C).

## Western Blot analysis and immuno-detection
Protein extracts dissolved in Lämmli sample buffer or SDS sample buffer were separated by SDS–PAGE (Biorad Mini Protean) and transferred to ethanol activated Amersham Hybond-P 0.45 µm PVDF membranes (Cytiva, Cat. #10600023) by either semi-dry or wet electro-blotting transfer buffer (25 mM Tris-HCl, 192 mM Glycine, 20% EtOH, 0.1% SDS). Membranes were blocked in 5% milk in TBS-T for 1 h at room temperature, and incubated with primary antibodies over night at 4 °C. Membranes were afterwards washed 5 – 10 times with TBS-T (50 mM Tris-HCl pH 7.5, 500 mM NaCl, 0.05% Tween-20). Secondary antibodies conjugated to HRP (horseradish peroxidase) were diluted in 5% Milk in TBS-T and added to the membranes for 60 – 90 min at room temperature, followed by 5 – 10 washing steps with TBS-T. Membranes were developed using ECL HRP substrate (Advansta, Cat. #541005X). Oxidized Luminol was detected using medical X-Ray films. Films were scanned and further processed with GIMP (version 2.10). No non-linear processing was applied. Antibodies used in this study are listed in Supplementary Table 2. The α-Dsc2, α-Dsc3, α-Tul1 and α-Ubx3 antibodies were kindly provided by Peter J. Espenshade, Johns Hopkins University School of Medicine.

## Immunodetection of ubiquitinated proteins
For detection of ubiquitinated proteins, samples were separated by SDS–PAGE gel electrophoresis using 12.5% gels. Proteins were then transferred to Amersham Hybond-P 0.45 µm PVDF membranes (Cytiva, Cat. #10600023) by wet blotting (80 V, 2 h) at 4 °C with transfer buffer (25 mM Tris-HCl, 192 mM Glycine, 20% EtOH, 0.1% SDS). After transfer, the membrane was blocked with 5% BSA in TBS-T (50 mM Tris-HCl pH 7.5, 150 mM NaCl, 0.45% Tween-20) before overnight incubation with P4D1 anti-ubiquitin antibody (Santa Cruz, Cat. # sc-8017) with 0.45% Tween-20 in 5% BSA blocking solution. Washing was done as described above. Membranes were developed using ECL select (Cytiva, Cat. # GERPN2235).

## Denaturing immunoprecipitation of ubiquitinated species
Denaturing immunoprecipitation of FLAG-Hmx1 and truncated versions of eGFP-ALFA-Hmx1 was adapted from (Schmidt et al[7].). Briefly, 50 $OD_{600}$ (150 $OD_{600}$ for downstream ubiquitin linkage immunoprecipitation) of logarithmically growing yeast were collected with 1 mL ice-cold 10 mM NaF. Cell pellets were resuspended in 0.1 M NaOH and incubated for 5 min at RT. After centrifugation (3 min, 21,000 x g, 4 °C), pellets were resuspended in 200 µL SDS lysis buffer (50 mM Tris-HCl pH 8, 1 mM EDTA, 1% SDS, 2 M urea, 10 mM NaF) supplemented with protease inhibitors (1x Complete® EDTA-free (Roche, Cat. # C7698), 1 mM PMSF, 10 mM N-ethyl maleinimid). Pellets were then solubilized by vortexing with glass beads (0.75–1 mm; 10 min at RT) and denatured (42 °C, 1 h). Lysates were diluted by addition of 1800 µL immunoprecipitation (IP) buffer (50 mM Tris-HCl pH 8, 100 mM NaCl, 1% Triton X-100, supplemented with protease inhibitors including yeast protease inhibitors (Sigma, Cat. # P8215) and cleared by centrifugation (10 min; 13,000 g; 4 °C). Supernatants were then added to prewashed anti-FLAG M2 magnetic beads (Sigma, Cat. # M8823) or anti-ALFA ST magnetic beads (NanoTag Biotechnologies, Cat. # N1516) and immunoprecipitated for 2 – 3 h at 4 °C. The beads were washed twice with IP buffer (50 mM Tris-HCl, pH 8, 100 mM NaCl, 1% Triton X-100) and thrice with wash buffer (50 mM Tris-HCl pH 8, 300 mM NaCl, 0.1% Tween-20) for 5 min at 4 °C. FLAG-tagged proteins were eluted by addition of 40 µL IP buffer supplemented with 500 ng/mL 3xFLAG peptide (Sigma, Cat. # F4799) and incubated for 1 h at room temperature. Magnetic beads were removed and the eluate was supplemented with 5x SDS sample buffer (250 mM Tris-HCl pH 6.8, 10% SDS, 50% glycerol, 10% β-mercaptoethanol) and denatured for 10 min at 95 °C. ALFA-tagged proteins were eluted with 50 µL 2x SDS sample buffer (100 mM Tris-HCl pH 6.8, 4% SDS, 20% glycerol, 4% β-mercaptoethanol) and denatured for 10 min at 95 °C.

For further immunoprecipitation with K48 or K63 selector resins, FLAG-proteins were eluted with 120 µL IP buffer as described above. ALFA-tagged proteins were immunoprecipitated with anti-ALFA CE agarose beads (NanoTag Biotechnologies, Cat. # N1512-L) instead of anti-ALFA ST and eluted twice with 60 µL IP buffer supplemented with 800 µg/mL ALFA peptide (NanoTag Biotechnologies) and incubated for 15 min at 37 °C. In both cases, eluates were split into three parts. The fraction for detection of holo-ubiquitinated species was denatured immediately after elution with 5x SDS sample buffer for 10 min at 95 °C. The remaining fraction was diluted with 720 µL IP buffer supplemented with protease inhibitors (1x Complete® EDTA-free (Roche, Cat. # C7698), 1 mM PMSF, 10 mM N-ethyl maleinimid, yeast protease inhibitors (Sigma, Cat. # P8215)) and equally split to anti-Ubiquitin K48 Selector agarose beads (NanoTag Biotechnologies, Cat. # N1810) and anti-Ubiquitin K63 Selector agarose beads (NanoTag Biotechnologies, Cat. # N1910). Beads were incubated for 1 h at 4 °C followed by two washes with IP buffer and three washes with Wash buffer. Ubiquitinated species were eluted and denatured with 2x SDS sample buffer for 10 min at 95 °C.

## Non-denaturing immunoprecipitation
Non-denaturing isolation of FLAG-Orm2 and Ubx3-FLAG was performed essentially as described before (Schmidt et al[7].) from 1000 $OD_{600}$ of cryo-ground cells. For non-denaturing isolation of FLAG-Hmx1, for truncated versions of eGFP-ALFA-Hmx1, Yif1-HA and for Ubx3-FLAG substrate interaction, logarithmically growing cells (100 $OD_{600}$ for Hmx1 and Ubx3-FLAG, 50 $OD_{600}$ for HA) were harvested by centrifugation with ice cold 10 mM NaF, and pellets were frozen in liquid nitrogen. Frozen pellets were resuspended in 500 µL lysis buffer (50 mM HEPES/KOH, pH 6.8; 150 mM KOAc, 2 mM Mg(OAc)₂; 1 mM CaCl₂; 15% glycerol) supplemented with protease inhibitors (1x Complete® EDTA-free (Roche, Cat. # C7698), 1 mM PMSF, 10 mM N-ethyl maleinimid, yeast protease inhibitors (Sigma, Cat. # P8215)) and solubilized by vortexing with glass beads (0.75–1 mm; 10 min at 4 °C).

Lysates were then diluted with 500 μL lysis buffer supplemented with protease inhibitors and 2% glyco-diosgenin (Sigma, Cat. # 850525) and incubated rotating at 4 °C for 1 h. Unsolubilized material was removed by centrifugation (13,000 × g; 15 min; 4 °C). The supernatant (800 μL) was added to equilibrated anti-FLAG M2 magnetic beads (Sigma, Cat. # M8823), anti-ALFA ST magnetic beads (NanoTag Biotechnologies, Cat. # N1516) or anti-HA magnetic beads (Thermo Fisher, Cat. #88836) and incubated for 2 – 3 h at 4 °C. Beads were recovered with a magnetic rack and washed two times with IP buffer containing 0.1% glyco-diosgenin and three times with IP buffer containing 0.1% glyco-diosgenin and 300 mM NaCl for 5 min at 4 °C. FLAG-proteins were eluted with 500 ng/mL 3xFLAG peptide (Sigma, Cat. # F4799) at RT for 1 h and supplemented with 5x SDS sample buffer (250 mM Tris-HCl pH 6.8, 10% SDS, 50% glycerol, 10% β-mercaptoethanol) followed by denaturation at 95 °C for 10 min. ALFA-tagged and HA-tagged proteins were eluted with 50 μL 2x SDS sample buffer (100 mM Tris-HCl pH 6.8, 4% SDS, 20% glycerol, 4% β-mercaptoethanol) and denatured for 10 min at 95 °C.

### Size exclusion chromatography

Non-denaturing immunoprecipitation of Ubx3-FLAG was performed as described above. Prior to elution, beads were additionally washed two times with PBS supplemented with 0.1% glyco-diosgenin (Sigma, Cat. # 850525). Bound material was eluted (1 h at RT, shaking at 850 rpm) from anti-FLAG M2 magnetic beads (Sigma, Cat. # M8823) with homemade 10 μg/μL Strep-6xFLAG-peptide in PBS supplemented with 0.1% glyco-diosgenin. After elution, FLAG-beads were separated with a magnetic rack and PBS-washed MagStrep"type3" XT Beads (IBA, Cat. #2-4090-002) were added to eliminate excess of Strep-6xFLAG-Peptide (1 h at RT, shaking at 850 rpm). The supernatant of this purification was used for size exclusion chromatography (SEC).

The SEC analysis was performed utilizing an ÄKTA Ettan LC High Performance Liquid Chromatography (HPLC) system coupled with a Superose 6 Increase 3.2/300 column. The column was pre-equilibrated with a mobile phase composed of PBS with 0.05% GDN (25x critical micelle concentration (CMC)). Calibration was executed using molecular weight standards, including Thyroglobulin (669 kDa), Ferritin (440 kDa), Aldolase (158 kDa), Ovalbumin (44 kDa), and Aprotinin (6.5 kDa). For sample analysis, 100 μl of eluate was loaded onto the ÄKTA Ettan LC system, employing the mobile phase at a flow rate of 0.034 mL/min. Fraction collection was conducted in 80 μL aliquots using a fraction collector. After collection, fractions were analyzed by SDS PAGE and Western Blot analysis.

### Live cell fluorescence microscopy

Live cell epifluorescence microscopy was carried out using a Zeiss Axio Imager M1 equipped with a SPOT Xplorer CCD camera, standard fluorescent filters and Visitron VisiView software (version 2.1.4). For Figs. 3g, 5e, 6e, f, 7h, and Supplementary Fig. 3d, Leica THUNDER Imager Live Cell (1111526683526683) equipped with a Leica K8 camera (11115471), a 100x/ 1.4 oil immersion objective (11506381), and Thunder 3D DCV Tera (111188922) was used. For computational clearing the Large V- Large Volume Computational Clearing (LVCC) option was used. For microscopy, cells were grown to midlog (OD$_{600}$ 0.5 – 0.8) phase in YNB media, concentrated by centrifugation, resuspended in A.d. and directly mounted onto glass slides. Brightness and contrast of the images in the figure were adjusted using ImageJ software (version 1.54 g).

To quantify the degradation of mNeonGreen (NG) -Hmx1 in the background of WT and tul1Δ cells before and after addition of CHX, fluorescence intensity of NG-Hmx1 was quantified using ImageJ software. Class E compartments (ROIs) were defined by mCherry-CPS signals and the mean fluorescence intensity of NG-Hmx1 within each ROI was measured. Background fluorescence was subtracted using a region outside the cells. The resulting values were depicted as a box-and-whisker plot, using GraphPad Prism. Statistical significance was determined using an unpaired t-test, with p < 0.05 considered significant.

### Cycloheximide chase assay

Logarithmically growing cells (20 OD$_{600}$) were harvested by centrifugation. Cells were resuspended in 50 mL fresh medium (for experiments with temperature-sensitive mutants the medium was pre-warmed at 30 °C). For chemical proteasome inhibition pdr5Δ cells were pre-incubated with 50 μM MG-132 (Sigma, Cat. # 474787) for 10 min. At t = 0 min 10 mL (4 OD$_{600}$) were harvested by centrifugation, washed once with ice-cold 10 mM NaF (see above), and pellets were snap frozen in liquid nitrogen. To the remaining culture 50 μg/mL CHX (Sigma, Cat. # C7698) was added from a 10 mg/mL stock. After the indicated time points 10 mL culture were harvested, washed and frozen as above. Whole cell extracts were prepared by alkaline extraction. SDS–PAGE, Western blot detection, and quantification were done as described above.

### Subcellular fractionation

The subcellular fractionation protocol was adapted from refs. 7,42,43. Briefly, logarithmically growing vps4Δ CDC48 or cdc48-3 cells, or vps4Δ ubx3$^{ΔUBX}$ cells, were shifted to 37 °C for 2 h and simultaneously treated with 50 μM MG-132 (Sigma, Cat. # 474787) or the respective vehicle. 40 OD$_{600}$ of midlog phase (OD600 = 0.5 – 1) growing cells were collected and resuspended in 1 mL ice-cold 10 mM NaF. Cell pellets were resuspended in 500 μL MF Buffer (300 mM Sorbitol, 1 mM EDTA, 20 mM Tris-HCl pH 7.5, 100 mM NaCl), supplemented with protease inhibitors (1x Complete® EDTA-free (Roche, Cat. # C7698), 1 mM PMSF, 10 mM N-ethyl maleinimid, yeast protease inhibitors (Sigma, Cat. # P8215)), 100 μL 0.75 – 1 mm glass beads were added and vortexed (6 ×1 min at 4 °C). Cell lysates were cleared at 2500 × g for 5 min. 400 μL of the supernatant was ultracentrifuged at 100,000 × g for 1 – 1.5 h to separate pellet (P100) and supernatant (S100) fractions. 300 μL of the S100 fraction were removed diluted with IP buffer (50 mM Tris-HCl pH 8, 100 mM NaCl, 1% Triton X-100) supplemented with protease inhibitors (1x Complete® EDTA-free (Roche, Cat. # C7698), 1 mM PMSF, 10 mM N-ethyl maleinimid, yeast protease inhibitors (Sigma, Cat. # P8215) and added directly to the anti-FLAG M2 magnetic beads (Sigma, Cat. # M8823). The P100 was washed once in cold MF Buffer without disturbing the pellet and centrifuged at 13,000×g for 5 min. The supernatant was discarded and the washed P100 fraction was resuspended in 200 μL SDS lysis buffer (50 mM Tris-HCl pH 8, 1 mM EDTA, 1% SDS, 2 M urea, 10 mM NaF) supplemented with protease inhibitors (1x Complete® EDTA-free (Roche, Cat. # C7698), 1 mM PMSF, 10 mM N-ethyl maleinimid). An equivalent amount of P100 (compared to S100) was transferred to IP buffer (50 mM Tris-HCl pH 8, 100 mM NaCl, 1.0% Triton X-100) supplemented with protease inhibitors (1x Complete® EDTA-free (Roche, Cat. # C7698), 1 mM PMSF, 10 mM N-ethyl maleinimid, yeast protease inhibitors (Sigma, Cat. # P8215)) and added to the anti-FLAG M2 magnetic beads (Sigma, Cat. # M8823). Immunoprecipitations under denaturing conditions were described above.

### Full lipidome analysis

Isogenic WT and mutant strains were grown into logarithmic phase in minimal selection medium. Approximately 30 OD$_{600}$ cells were rapidly harvested and frozen in liquid nitrogen. Pellets were resuspended in 1.5 mL sterile, ice-cold water and the OD$_{600}$ was measured. Exactly 20 OD cells were transferred into a fresh tube and water added to a final volume of 1 mL. 400 μL 0.5 mm glass beads were added, and glass bead lysis was performed for 15 min at 4 °C. 750 μL of the lysate were transferred to fresh tubes, frozen in liquid nitrogen, and shipped to Lipotype (Dresden, Germany) for subsequent lipid extraction and shotgun mass spectrometric lipidomics analysis. Lipidomic profiling was conducted across four independent experiments. Quantitative

data is usually displayed as mean ± standard deviation from at least 3 biological replicates. Statistical significance was tested in Prism software (version 10.1.2) by multiple t-tests correcting for multiple comparisons[71], with a false discovery rate Q = 1%, without assuming individual variance for each sample. *P*-values were flagged with > 0.05 (ns), ≤ 0.05 (*), ≤ 0.01 (**) and ≤ 0.001 (***).

## ß-Estradiol induced protein expression

To induce protein expression, 500 nM β-Estradiol (Sigma, Cat. #E2758) was added to logarithmically grown overnight cell cultures. Protein induction and/or localization was examined at the indicated time points using Western blot analysis and/or live cell fluorescent microscopy. To examine protein degradation kinetics, 4 $OD_{600}$ of cells were harvested after 90 min of β-Estradiol induction (t = 0) by centrifugation and snap frozen in liquid nitrogen. To the remaining culture 50 μg/mL cycloheximide (Sigma, Cat. # C7698) was added. At the indicated time points 4 $OD_{600}$ of culture were harvested, and snap frozen in liquid nitrogen. Whole cell extracts were prepared by alkaline extraction. SDS−PAGE, Western blot detection, and quantification were done as described above.

## Quantification of Western Blot analysis

Western blot signals were quantified by densitometry using ImageJ software (version 1.54 g). Quantifications were exported to Microsoft Excel (Version 16.0.5422.1000), normalized to the respective Pgk1 loading controls, and presented as mean ± standard deviation of the mean from at least three independent experiments. t = 0 was set to 1.

## Reporting summary

Further information on research design is available in the Nature Portfolio Reporting Summary linked to this article.

## Data availability

All relevant data has been included in the paper in the main figures and in supplementary data 1–4, supplementary tables 1 and 2, and figures. The proteomics data have been deposited in PRIDE (accession number: PXD055553). The original trajectories for the MD simulations have been deposited in Zenodo and are available under 10.5281/zenodo.13889274 and 10.5281/zenodo.13889576. All other data that support the findings of this study are available from the corresponding author upon request. Source data are provided with this paper.

## Code availability

The custom-made R script that was generated to isolate and align predicted TMDs with respect to their bilayer orientation will be made available upon request.

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

## Acknowledgements

We thank Snezhana Oliferenko, Hesso Farhan, Chris Dunworth, and Lukas A Huber for critically reading the manuscript, Ming Li, Peter

Espenshade, Sebastien Leon, and Scott Emr for reagents, Bob Kaufmann for help in characterizing the Dsc2 L1 loop mutant. This research was funded in part by the Austrian Science Fund (FWF) (10.55776/P32161, 10.55776/P34907, 10.55776/DOC82 to DT, and 10.55776/P36187 to OS), by a Lipotype lipidomics excellence award (LEA 2019) to OS, by a Luxembourg National Research Fund (FNR): Grant #13571826 to YW, and by European Union's Horizon 2020 research and innovation program under the Marie Skłodowska-Curie grant agreement No. 847681 (to KRL). For open access purposes, the author has applied a CC BY public copyright license to any author accepted manuscript version arising from this submission.

## Author contributions

YW, SIS, designed and performed the vast majority of experiments and analyzed data, with support from AP, KS, AR, TDM, MAW, VN, NJMM, MGT, SW. BS and LK conducted quantitative proteomics analysis, OS prepared samples for lipidomics analysis. XT and KR performed and analyzed MDS. DT and YW wrote the manuscript. DT conceived and supervised the study.

## Competing interests

The authors declare no competing interests.
