## [Transparent Peer Review file · Nature Communications]

The Dsc ubiquitin ligase complex identifies transmembrane degrons to degrade orphaned proteins at the Golgi

Corresponding Author: Professor David Teis

Version 0:

Reviewer comments:

Reviewer #1

(Remarks to the Author)

The authors assessed *tul1 vps4* mutant *S. cerevisiae* and plasmid-rescued WT cells after growth in [¹³C6553 ¹⁵N₂]-L-lysine for plasmid-rescued WT cells or [¹²C6555 ¹⁴N₂]-L-lysine for deletion mutant cells for SILAC proteomics of tryptic digests followed by LC-MS/MS for n=3 biological replicates. For *tul1* cells, selected up-regulated proteins are indicated and separately for *vps4* mutant cells in Fig1 as volcano plots for total proteins and membrane proteins. The authors report the hydrophobic lengths of transmembrane domains for the membrane proteins up-regulated in *tul1 vps4* mutant cells.

Glass bead lysis of *tul1 gld1Δ* mutant, *Om2KR* cells, and WT cells for lipidomic analysis by Lipotype Lipidomics company for mass spectrometry and reported in Figure 2 for PI, PE, and PC species characterized.

The authors expressed 3XFLAG-HMX1 in WT cells, *tul1Δ*, and *dsc2Δ* mutants followed by anti-FLAG magnetic bead isolation from supernatants of 1%SDS lysates vortexed with glass beads diluted with 1%Triton-X-100 and cleared by centrifugation. Elution by 3XFLAG peptide and magnetic bead removal followed by SDS PAGE and western blotting for anti-ubiquitin was indicated for expressed 3XFLAG-HMX1 in WT cells but not *tul1Δ*, and *dsc2Δ* mutants. Co-IP of FLAG-Hmx1 with *Tul1*, *Dsc2*, *Dsc3*, and *Ubx3* is also indicated for WT cells. Quantification of western blots in Figure 3 for FLAG-Hmx1 is indicated in cell lysates at 0, 90, 180 min after CHX addition to WT, *tul1Δ*, and *dsc2Δ* mutants, as well as MG-132 and *vps4* mutant cells. The authors indicate in Figure 3 the expression of β-estradiol inducible NG-Hmx1 visualized by live cell epi-fluorescence for WT and *tul1Δ* mutant cells for mCherry-Cps1 and NG-Hmx1 visualized at 40 and 120 min after β-estradiol. In Figure 4, western blots for Ng-Hmx1 in lysates of *vps4* mutant cells and *tul1Δ vps4* double mutant cells are indicated after incubation of cells 0, 90, 180 minutes of CHX and visualized by live cell epi-fluorescence for mCherry-Cps1 and NG-Hmx1 after cell incubation for 180 min with CHX.

The authors report western blots of anti-ubiquitin after IP of expressed FLAG-Hmx1 *vps4* mutant or *tul1Δ vps4* double mutant or *tul1Δ* cells, and a second IP with nanobodies to either K63 or K48 polyubiquitin. The authors studied the 100,000g pellet and supernatant from homogenized *vps4* mutant cells expressing FLAG-Hmx1 and CDC48 or *cdc-48-3* mutant (in a *pdr5Δ vps4Δ* background) previously incubated with MG-132 with western blots for anti-ubiquitin after FLAG-Hmx1 IP. The authors indicate in Figure 4 *ubx3* UBX cells and *vps4-EQ* cells incubated with cycloheximide for 0,90, and 180 min with western blots of total cell extracts for anti-FLAG-Hmx1 with experiments also indicated for β-estradiol inducible NG-Hmx1. The authors expressed *Yif1-3xHA* in WT or *tul1Δ* mutant cells followed by anti-HA IP and indicated western blots for *Tul1*, *Ubx3*, *Dsc3*, *Dsc2*, and *YIF1-HA*. Western blots of anti-HA for WT or *tul1Δ* mutant cells expressing *YIF1-HA* incubated with MG-132 and Cycloheximide are shown. Westerns are indicated for anti-HA in *YIF1-HA* expressing cells grown at 30 degrees in *cdc-48-3* or *tul1Δ* mutant cells at 0, 120, 240 min of cycloheximide addition and at 0,180, 360 min of cycloheximide addition at the non-restrictive temperature for WT or *Tul1IRING* mutant or *ubx3* UBX mutant cells. Live cell epi-fluorescence is shown for *yif1Δ* cells expressing GFP-Yif1 with *Mup1-mCherry*.

The authors report co-IP experiments of cells expressing FLAG-Hmx1, FLAG-*Om2-KR* or *YIF-HA* in WT, *ubx3* UBX, *tul1Δ*, *dsc2Δ*, or *dsc3Δ* mutant cells followed by Western blotting for *Tul1*, *Ubx3*, *Dsc3*, or *Dsc2*. The authors show the eluate of 13 fractions from size exclusion chromatography of eluted *Ubx3*, *Tul1*, *Dsc3*, and *Dsc2* from *Ubx-3* IP from WT or *dsc3Δ* mutant cells and live cell epi-fluorescence for WT and mutant cells for *Ubx3-NG* and *dsRed-HDEL*.

The authors show the results of rhomboid like domain of *Dsc2* embedded in indicated lipids with molecular dynamics assessed after stimulation for 100ns. As well, *Hmx1* from residues 264-371 as well as with an extension with additional LIV residues or LIVLIV amino acid residues N terminally fused to mNeonGreen-ALFA-(NG)-tags or GFP-ALFA-(GFP) expressed from the *TDH3* promoter. Western blots of NG-Hmx1 are indicated after incubation for 0,60, 120 min in cycloheximide. incubated for 0,30,60 min of cycloheximide with anti-GFP westerns is indicated. IP of GFP in *tul1Δ* mutant cells followed by

western blots of Dsc3, Dsc2, and Ubx3 are shown. Live cell epifluorescence of the GFP Hmx1 construct of residues 264-371 and FM-64 is shown. IP of the construct in *tul1Δ* mutant cells or KR cells with anti-ubiquitin western blotting is shown under denaturing IP or non-denaturing IP conditions with the latter including *ubx3* UBX, *tul1Δ*, *dsc2Δ*, or *dsc3Δ* mutant cells and westerns for Tul1, Ubx3, Dsc3, and Dsc2.

From the above and additional data in the Supplementary Figures and supplementary tables, the authors conclude that they have demonstrated the role of the Dsc3 ubiquitin ligase in selecting orphaned proteins at the Golgi for ubiquitination and degradation. However, this conclusion may have already been made in the authors' previous study in ref 7. What may be an advance is the last paragraph in the Discussion where the conclusion is made that protein quality control for orphaned membrane proteins is via Golgi-located monitoring of transmembrane domain length of orphaned membrane proteins.

The coherence of the Results may need to be reconsidered. For the SILAC experiments, it is unclear if the strategy has influenced the experiments indicated above. For example, Hmx1 is not found in the total proteome and has few peptides in the membrane proteome. Also featured in the extensive experiments is Yif1 not detected in any of the SILAC experiments. The authors have referred to Hmx1 in their prior work in ref 7 rendering the SILAC experiments perhaps not essential to the paper. It is recommended that the SILAC work be selected for another paper elsewhere. In any case, for the proteomics experiments, the authors may need to indicate if the raw data have been submitted to a public repository. The supplementary tables may need additional legends. Have the authors assessed if the same conclusions are deduced when the isotopes are reversed? For the lipidomics experiments again the same criticism may be considered of coherence with the paper and the lipids selected for the molecular dynamics experiments. The lipidomics section may be considered for another article submitted elsewhere.

For all Figures and supplementary figures have $n=3$ biological replicates been done for each experiment and can the westerns, coIP and live cell epifluorescence be quantified with $n=3$ biological replicates as well?

The coherence of the experimental design to the conclusion of the Dsc3 ubiquitin ligase in selecting orphaned proteins at the Golgi for ubiquitination and degradation may also need consideration. For example, what experiments in this work extend the authors' prior study in ref 7 that the ubiquitination is in the Golgi? Are the markers for these Golgi experiments in the current submitted manuscript sufficiently justified for this conclusion? The epifluorescence in ref 7 with markers chosen may be clearer than those in the current manuscript. For the conclusion of a Golgi-located quality control machinery monitoring transmembrane domain length for orphaned membrane proteins, the authors may consider a focused paper on this conclusion.

The authors may consider a clearly defined, logical, coherent, experimental strategy and data with $n=3$ biological replicates and quantification for all experiments.

Reviewer #2

(Remarks to the Author)

The manuscript submitted to Nature Communications by Teis and colleagues reports first on a characterization of the substrates, effects, and transmembrane domains (TMDs) that are regulated by Endosome and Golgi Associated Degradation (EGAD), a pathway that exhibits several ERAD-like components and features, including an E3 ubiquitin ligase complex. To define the substrates targeted by this ligase, known as Tul1, a proteome-wide screen was conducted using SILAC in wild type and *Tul1* delete cells, and the results were compared in an ESCRT-deficient strain, i.e., *vps4*. The efficacy of the *Tul1* screen was verified by the identification of ~50% of all TMD proteins as well as a known substrate, *Orm2*, and amongst the substrates it was striking that 21 were known ER/Golgi proteins and 44% possess short (e.g., <18 aa) TMDs; ~66% had at least one TMD <16 aa. Although there was some overlap in the *tul1* and *vps4* screen, amongst the 72 proteins in the latter screen, they were generally longer. Consistent with the regulation of lipids accorded by *Orm2* (and prior work from the Teis lab), PC and PE levels rose in the mutant as well as in a *gld1* mutant, which directs the complex to the Golgi. There was also an apparent increase in short/asymmetric lipids. Interestingly, one new EGAD substrate identified in this study—which is also targeted by ESCRT—is a conserved heme oxidase, Hmx1, and specific assays confirmed combined ubiquitin-proteasome- and ESCRT-dependent degradation. Based on these data, the authors examined Hmx1 residence after beta-estradiol induction: The protein resided in the class E/limiting vacuole membrane and remained at this location after CHX addition in a *tul1vps4* double mutant. A role for Cdc48, along with the Ubx3 UBX domain, in degradation was also clearly shown, and protein retrotranslocation was suggested (but see below). Next, the focus switched to an analysis of Yif1, which was absent from the proteomic screen but is a known Dsc/ESCRT substrate. As anticipated, protein turnover required Tul1, Cdc48, and Ubx3 activities, and as also expected the protein resided in the Golgi. Yet, overexpression freed the protein from the Golgi for ESCRT-dependent vacuolar degradation. The focus of the manuscript next turned to data consistent with the notion that the UBA-containing Dsc2 rhomboid-like iRhom protein is a recognition factor, or more appropriately is at least required for recognition of three substrates: Hmx1, a stabilized form of *Orm2*, and Yif1 since co-IP analyses in a *dsc2* mutant significantly decreased, to varying extents, association with the Dsc components, and most notably Tul1. Deletion of Ubx3 gave similar results, with the exception of one substrate. Nevertheless, these results are consistent with the function of iRhoms in other systems, along with the importance of Cdc48 recognition via a Ubx protein. Finally, MD simulations were used to suggest that Dsc2, like several other iRhoms, thins the bilayer, which may facilitate membrane extraction, and the regions within Hmx1—since it contains only a single TMD—that act as a putative 'degron' for Dsc2 were examined. Ultimately, the authors show that adding 1 or 2 additional alpha-helical turns progressively increases the stability of a simplified version of Hmx1 and that substrate interaction with Dsc2 and Ubx3 drops. Curiously, the Hmx1

substrate used here was preferentially K-48 ubiquitinated.

Overall, several quite intriguing and important results are presented in this study, yet some conclusions are overstated, which could be rectified by the application of more refined technologies. Other results are in line with what might be expected based on studies in related systems. Thus, there are questions about the manuscript's novelty, at least as it currently stands.

Comments:

1. While the authors focus on the 2-fold increase in PI-C-26 species, the effect on PE in Fig 2 may be just as profound. Is there a statistical difference between the effects on PI and PE? Is the same conclusion true for PC?
2. Is the integrated FLAG-Hmx1 protein fully active, as determined in growth assays for the function of this protein?
3. The statement, "loss of ESCRT function did not block Hmx1 degradation..." (p. 7) should be altered to the "loss of ESCRT function [alone] did not block...". In addition, this phenomenon is not uncommon, and is seen for ERAD substrates that escape the ER and are degraded in an ESCRT-dependent fashion.
4. The results with the K63 and K48 nanobodies are intriguing, but confidence for the authors' conclusions would be strengthened if the resulting elutes were blotted for antibodies specific for K48 and K63. Moreover, since mixed chains with K11 have been uncovered as being preferentially targeted for proteasome-dependent degradation, an analysis of this species should also be performed.
5. The quality of the blot shown in Fig 4D is poor. Replicates of this important experiment should be shown, along with simple quantification of whole band densities. It is also expected that ubiquitinated Hmx1 should be preferentially extracted into the S100 fraction. Is this true?
6. The importance of the vps4-EQ mutant is not discussed in the text (Fig. 4F), and data to fully support the statement that, "K48 polyubiquitination [is needed] for EGAD" are incomplete.
7. The logic underlying the tangential focus on Yip1 is unclear. Does the protein have short TMDs? Amongst other Dsc/ESCRT substrates, are there unique features that make Yip1 particularly worthy of study? If not, the data in Fig 5 could be included as a supplement and referred to in the context of the substrate recognition section.
8. The results in Fig 6 need to be repeated for statistical validation. Does removal of the UBX domain in Ubx3 give a null phenotype in this assay? Do we know which regions in Dsc2 are required specifically for 'recognition'? Does the loss of Dsc2 prevent retrotranslocation (viz Fig 4D)? While the results are not unexpected, more refined tools (such as site-specific crosslinking, which could show direct recognition of the TMD) might excitingly establish that Dsc2 is the prime recognition factor. This result that would substantially increase the novelty of the study.
9. The data in Fig S6 suggest variable Tul1 loading for several of the strains. This emphasizes the need for replicates of the co-IP experiments and necessitates a quantification of the relative amount of protein precipitated (divided by the input) in the various backgrounds used.
10. Studies to identify the Hmx1 degron for Dsc2 in Fig 7 require confidence that Dsc2 is indeed the direct recognition factor (see above).
11. In Fig 7E, do the levels of Tul1 interaction also decline?
12. In the absence of a live-cell ER marker, it is unclear if the protein localizes to the ER based simply on the images shown in Fig 7G and in Fig S7G.
13. The definition of a degron requires that a motif is completely transposable to another protein substrate and that the requirements for degradation are maintained. It does not appear that this condition was met.

Version 1:

Reviewer comments:

Reviewer #1

(Remarks to the Author)

The authors report SILAC labelling proteomics to report proteins in wt, tul1 gld1 , vld1 cells. Proteins increasing in abundance in the gene deleted cells were reported with predicted transmembrane lengths. Lipidomics is reported for wt, tul1 gld1 , Orm2KR mutant cells.

Expressed FLAG tagged Hmx1 IP was concluded to be polyubiquitinated based laddering in western blots with cycloheximide addition to cells expressing FLAG-Hmx1 concluded to lead to degradation but not in tul1 , dsc2 , ubx3 , dsc3 or gld1 cells. The effect of MG-132 is reported in wt and vps4 cells. Replacing Hmx1 with β -estradiol241 inducible mNeonGreen-ALFA-Hmx1 is reported with imaging for wt cells (replacing Hmx1) tul1 cells, tul1 rsp5G747E or rsp5G747E cells. Replacing Hmx1 with β -estradiol241 inducible mNeonGreen-ALFA-Hmx1 and expressing mCherry-Cps1 is reported as well in vps4 cells and tul1 vps4 cells.

FLAG-Hmx1 expressing wt, or vps4 or tul1 vps4 or tul1 or Vps4EQ and Ub-K63R, or Ub-K48R or Ub-K11R cells were assessed with antiK48ubiquitin antibodies and nanobodies.

FLAG-Hmx1 IP from S100 or P100 fractions from vps4 or vps4 cdc48-3 cells with or without MG-132 is reported as well as ubx3 UBX cells and Vps4EQ mutant and vps4D cells and β -estradiol241 inducible mNeonGreen-ALFA-Hmx1 cells.

The authors report IP studies with Orm2, Yif1 in FLAG-Hmx1 expressing cells deleted or not for indicated genes as well as

cells expressing Ubx3-FLAG as well as imaging of wt and gene deleted cells expressing Ubx3-Neon Green and dsRed-HDEL, Vps4-mCHERRY, Sec7-mCHERRY, Mnn9-BFP.

Molecular dynamic simulations of the rhomboid like domain of Dsc2 or Dsc2Y53A,R54A in 45% POPC, 45% POPE and 10% cholesterol with CHARMM-GUI simulated with indicated lipids is reported. Experiments with mutant Dsc2Y53A,R54A are reported in cells for GFP-Orm2 and NeonGreen-Hmx1 imaging.

Expression of indicated Hmx1 mutant constructs N-terminally fused to mNeonGreen-ALFA-(NG)-tags or GFP-ALFA-(GFP)-tags with western blots of cells with beta-estradiol or cycloheximide. Imaging of expressed GFP- Hmx1 is indicated and denaturing and non- denaturing IP.

The authors conclude a Dsc quality control at the Golgi for orphaned proteins with short transmembrane domains that they also conclude are degrons. Dsc2 thinning of membranes is reported with associated Ubx3 enabling cdc48 extraction to proteasome degradation of the selected orphaned proteins and Tul1 ubiquitination concluded from the studies.

Again, this reviewer could not find HMX1 in supplementary Table 1 tab WT-tul1d although it is in the other 2 tabs, wt-vps4d , vld1d-gld1d. This makes it confusing for Figure 1b for example. There is no mention of the SILAC proteomics conclusions in the Discussion that remain focussed on the proteins indicated in Figure 7i with HMX1 absent from supplementary Table 1 tab wt-tul1d. The authors have also indicated that Yif1 is absent from the SILAC proteomics. The paper remains an extension of their prior studies on HMX1 and Yif1 and if focussed on the claims of the discussion without the proteomics may be more coherent. Again for the lipidomics, did this lead directly to the design of the molecular dynamic simulations? Is the lipidomics needed or of archival value ?

An argument can be made for Figure 2 but the missing HMX1 in supplementary table 1 remains confusing.

The figures have been improved. A minor point is that the authors may consider SD rather than SEM for the figures. In Supplementary Information there does not seem to be any legends to describe the 3 supplementary movies.

Reviewer #2

(Remarks to the Author)

The authors have satisfactorily addressed the concerns of this reviewer.

Minor issues:

1. The 3rd column in Fig 5e is not labeled. Is this "dsRED-HDEL"?
2. Fig 7g legend doesn't refer to "dsc3delta cells", nor does the figure, which is what is what appears to be stated in the text.

Reviewer #1

Summary of our manuscript & comment 1: The authors assessed *tul1Δ*, *vps4Δ* mutant *S. cerevisiae* and plasmid-rescued WT cells after growth in [¹³C₆⁵⁵ ¹⁵N₂]-L-lysine for plasmid-rescued WT cells or [¹²C₆⁵⁵ ¹⁴N₂]-L-lysine for deletion mutant cells for SILAC proteomics of tryptic digests followed by LC-MS/MS for n=3 biological replicates. For *tul1Δ* cells, selected up-regulated proteins are indicated and separately for *vps4Δ* mutant cells in Fig1 as volcano plots for total proteins and membrane proteins. The authors report the hydrophobic lengths of transmembrane domains for the membrane proteins up-regulated in *tul1Δ*, *vps4Δ* mutant cells.

Glass bead lysis of *tul1Δ*, *gld1Δ* mutant, *Orm2KR* cells, and WT cells for lipidomic analysis by Lipotype Lipidomics company for mass spectrometry and reported in Figure 2 for PI, PE, and PC species characterized.

The authors expressed 3XFLAG-HMX1 in WT cells, *tul1Δ*, and *dsc2Δ* mutants followed by anti-FLAG magnetic bead isolation from supernatants of 1%SDS lysates vortexed with glass beads diluted with 1%Triton-X-100 and cleared by centrifugation. Elution by 3XFLAG peptide and magnetic bead removal followed by SDS PAGE and western blotting for anti-ubiquitin was indicated for expressed 3XFLAG-HMX1 in WT cells but not *tul1Δ*, and *dsc2Δ* mutants. Co-IP of FLAG-Hmx1 with Tul1, Dsc2, Dsc3, and Ubx3 is also indicated for WT cells. Quantification of western blots in Figure 3 for FLAG-Hmx1 is indicated in cell lysates at 0, 90, 180 min after CHX addition to WT, *tul1Δ*, and *dsc2Δ* mutants, as well as MG-132 and *vps4Δ* mutant cells. The authors indicate in Figure 3 the expression of β-estradiol inducible NG-Hmx1 visualized by live cell epi-fluorescence for WT and *tul1Δ* mutant cells for mCherry-Cps1 and NG-Hmx1 visualized at 40 and 120 min after β-estradiol. In Figure 4, western blots for Ng-Hmx1 in lysates of *vps4Δ* mutant cells and *tul1Δ vps4Δ* double mutant cells are indicated after incubation of cells 0, 90, 180 minutes of CHX and visualized by live cell epi-fluorescence for mCherry-Cps1 and NG-Hmx1 after cell incubation for 180 min with CHX. The authors report western blots of anti-ubiquitin after IP of expressed FLAG-Hmx1 *vps4Δ* mutant or *tul1Δ vps4Δ* double mutant or *tul1Δ* cells, and a second IP with nanobodies to either K63 or K48 polyubiquitin. The authors studied the 100,000g pellet and supernatant from homogenized *vps4Δ* mutant cells expressing FLAG-Hmx1 and CDC48 or *cdc-48-3* mutant (in a *pdr5Δ vps4Δ* background) previously incubated with MG-132 with western blots for anti-ubiquitin after FLAG-Hmx1 IP. The authors indicate in Figure 4 *ubx3ΔUBX* cells and *vps4-EQ* cells incubated with cycloheximide for 0,90, and 180 min with western blots of total cell extracts for anti-FLAG-Hmx1 with experiments also indicated for β-estradiol inducible NG-Hmx1.

The authors expressed Yif1-3xHA in WT or *tul1Δ* mutant cells followed by anti-HA IP and indicated western blots for Tul1, Ubx3, Dsc3, Dsc2, and YIF1-HA. Western blots of anti-HA for WT or *tul1Δ* mutant cells expressing YIF1-HA incubated with MG-132 and Cycloheximide

are shown. Westerns are indicated for anti-HA in YIF1-HA expressing cells grown at 30 degrees in *cdc-48-3* or *tul1Δ* mutant cells at 0, 120, 240 min of cycloheximide addition and at 0,180, 360 min of cycloheximide addition at the non-restrictive temperature for WT or *Tul1RING* mutant or *ubx3ΔUBX* mutant cells. Live cell epifluorescence is shown for *yif1Δ* cells expressing GFP-Yif1 with Mup1-mCherry.

The authors report co-IP experiments of cells expressing FLAG-Hmx1, FLAG-Orm2-KR or YIF-HA in WT, *ubx3ΔUBX*, *tul1Δ*, *dsc2Δ*, or *dsc3Δ* mutant cells followed by Western blotting for Tul1, Ubx3, Dsc3, or Dsc2. The authors show the eluate of 13 fractions from size exclusion chromatography of eluted Ubx3, Tul1, Dsc3, and Dsc2 from Ubx-3 IP from WT or *dsc3Δ* mutant cells and live cell epifluorescence for WT and mutant cells for Ubx3-NG and dsRed-HDEL.

The authors show the results of rhomboid like domain of Dsc2 embedded in indicated lipids with molecular dynamics assessed after stimulation for 100ns. As well, Hmx1 from residues 264-371 as well as with an extension with additional LIV residues or LIVLIV amino acid residues N terminally fused to mNeonGreen-ALFA-(NG)-tags or GFP-ALFA-(GFP) expressed from the TDH3 promotor. Western blots of NG-Hmx1 are indicated after incubation for 0,60, 120 min in cycloheximide. incubated for 0,30,60 min of cycloheximide with anti-GFP westerns is indicated. IP of GFP in *tul1Δ* mutant cells followed by western blots of DSc3, Dsc2, and Ubx3 are shown. Live cell epifluorescence of the GFP Hmx1 construct of residues 264-371 and FM-64 is shown. IP of the construct in *tul1Δ* mutant cells or KR cells with anti-ubiquitin western blotting is shown under denaturing IP or non -non-denaturing IP conditions with the latter including *ubx3ΔUBX*, *tul1Δ*, *dsc2Δ*, or *dsc3Δ* mutant cells and westerns for Tul1, Ubx3, Dsc3, and Dsc2.

From the above and additional data in the Supplementary Figures and supplementary tables, the authors conclude that they have demonstrated the role of the Dsc3 ubiquitin ligase in selecting orphaned proteins at the Golgi for ubiquitination and degradation. However, this conclusion may have already been made in the authors' previous study in ref 7. What may be an advance is the last paragraph in the Discussion where the conclusion is made that protein quality control for orphaned membrane proteins is via Golgi-located monitoring of transmembrane domain length of orphaned membrane proteins.

Answer: We thank the reviewer for the thorough evaluation of our manuscript and for the insightful, constructive comments. The lengthy revision process was, in part, caused by lab accident that started a fire and devastated our equipment room and by an accident of the first author, who sliced through his middle finger, while cutting up a melon. Therefore, we recruited additional scientists, who are now new co-authors, to support the revision process.

Our earlier paper (ref. 7 Schmidt et al. 2019) reported the discovery that the Dsc complex mediates the endosome and Golgi associated proteasomal degradation (EGAD) of Orm2 and thereby controls sphingolipid metabolism. This was an important finding, but with limitations,

since we only identified and characterized a single endogenous substrate of the Dsc complex (Orm2). It has remained unclear (1) how the Dsc complex detects its substrates and (2) if the Dsc complex has a general function in Golgi quality control.

Now we have closed these knowledge gaps. We **expand the substrate spectrum of the Dsc complex** with Hmx1, and Yif1, and **we show that the Dsc complex detects short transmembrane domains in orphaned proteins via Dsc2** and targets them for **EGAD (Orm2, Yif1, Hmx1) or ESCRT (Hmx1)** mediated degradation. These results could not have been predicted by our earlier work, and establish how the Dsc complex contributes to Golgi quality control and to cellular proteostasis in general.

Comment 2: The coherence of the Results may need to be reconsidered:

For the SILAC experiments, it is unclear if the strategy has influenced the experiments indicated above. For example, Hmx1 is not found in the total proteome and has few peptides in the membrane proteome. Also featured in the extensive experiments is Yif1 not detected in any of the SILAC experiments. The authors have referred to Hmx1 in their prior work in ref 7 rendering the SILAC experiments perhaps not essential to the paper. It is recommended that the SILAC work be selected for another paper elsewhere. In any case, for the proteomics experiments, the authors may need to indicate if the raw data have been submitted to a public repository. The supplementary tables may need additional legends. Have the authors assessed if the same conclusions are deduced when the isotopes are reversed?

Answer: To address the reviewers concern regarding isotope labeling, we conducted additional SILAC experiments and compared the proteomes of *vld1Δ* mutants (which cannot target the Dsc complex to the limiting membrane of vacuoles; labeled with heavy lysine) to *gld1Δ* mutants (which cannot target the Dsc complex to the Golgi; labeled with light lysine). The new results show that in *gld1Δ* mutants a similar set of membrane proteins accumulated compared to *tul1Δ* mutants (**new Supplementary Figure 1, Supplementary tables 1, 2**). Hence the Golgi resident Gld1-Dsc complex, but not the vacuolar Vld1-Dsc complex controls the protein levels of Hmx1, Orm2, Csg2 and of several other membrane proteins. We also would like to emphasize that all strains in this study are genetically and metabolically isogenic. Based on the new results, we conclude (a) that the labeling strategy does not influence the degradation and the identification of Dsc complex substrates and (b) that the Golgi resident Gld1-Dsc1 is required for protein quality control.

The reviewer commented that ‘The authors have referred to Hmx1 in their prior work in ref 7 rendering the SILAC experiments perhaps not essential to the paper. It is recommended that the SILAC work be selected for another paper elsewhere’. The reviewer is correct that Hmx1 was identified in our earlier work (ref 7). In this earlier work, we used SILAC pulse chase experiments to compare protein degradation in cells that lacked the vacuolar master-protease Pep4. Therefore, our previous work purposely ignored vacuolar degradation, which led to the

identification of Orm2 as a Dsc complex substrate that was degraded by the proteasome via EGAD.

Our new SILAC experiments (**Figure 1, new Supplementary Figure 1, Supplementary tables 1, 2**) include Dsc substrates that are degraded in vacuoles. Therefore, we now identify substrates of the Golgi resident Gld1-Dsc complex that are degraded by EGAD or ESCRT pathways, or both (such as Hmx1). We believe that these findings are quite remarkable. Moreover, we have now used a novel mass spectrometer (Orbitrap Eclipse), which increased protein quantification by > 30%. In the 2019 EMBO paper, we quantified 2411 proteins, now we have quantified > 3600 proteins.

We respectfully disagree with this comment: ‘For example, Hmx1 is not found in the total proteome and has few peptides in the membrane proteome.’ Hmx1 was detected in every quantitative proteomics experiment (including the new experiment comparing *vld1Δ* to *gld1Δ*) in the total proteome with several H/L peptide pairs (in total ≥ 15 H/L pairs). In WT/*vps4Δ*, in WT/*tul1Δ*, and in *vld1Δ/gld1Δ* the H/L ratio was similar in each the individual biological replicate with 0,342; 0,316; 0,368 or 0,448; 0,449; 0,417, or 0,334; 0,202; 0,353 respectively, with p values of $3,92 \times 10^{-16}$ or $4,28 \times 10^{-16}$, or 1×10^{-17} demonstrating excellent reproducibility (see **Supplementary tables 1, 2**).

The reviewer is correct: ‘Also featured in the extensive experiments is Yif1 not detected in any of the SILAC experiments’. We chose to examine Yif1, (1) because Yif1 was not represented in the quantitative proteome analysis, (2) because Yif1 also features a short TMD1 (18aa) and (3) because Yif1 had been described to be a Dsc complex substrate that is degraded by the ESCRT pathway. Yet, since also reviewer 2 made a similar point, we substantially shortened the part on Yif1, and moved the remaining results to **Supplementary Figure 5a, b**.

We have submitted the proteomics data to PRIDE (**accession number: PXD055553**). Also, we have added headers and additional legends to the supplementary tables.

Comment 3: For the lipidomics experiments again the same criticism may be considered of coherence with the paper and the lipids selected for the molecular dynamics experiments. The lipidomics section may be considered for another article submitted elsewhere.

Answer: We thank the reviewer for the excellent suggestion to include asymmetric lipids in the MD simulation. We have now used asymmetric lipids PC(10:0;16:0) or PC(10:1;16:0) in MD simulation with Dsc2. We observed that the saturated C10 intermediate acyl chain stably associated with Dsc2 (**new Supplementary Figure 6b, c**), whereas unsaturated ones (C10:1), or symmetric lipids (POPC) did not stably interact with Dsc2.

Hence, the lipidomics experiments complement the proteomic data and together with new MD modeling indicate for the first time a functional relationship of asymmetric lipids, short TMDs and Golgi protein quality by the Dsc complex.

Certainly, additional experiments are required to understand the role of asymmetric lipids and membrane protein quality control, but these experiments go beyond the scope of our current study.

Comment 4: For all figures and supplementary figures have n=3 biological replicates been done for each experiment and can the westerns, coIP and live cell epifluorescence be quantified with n-3 biological replicates as well?

Answer: Yes, each experiment in the manuscript was performed at least three times. The quantifications of key western blot and live cell imaging experiments are provided throughout the manuscript (**Figures 3f, g, Supplementary Figure 3a, e, Figure 4b, e, Supplementary Figure 4a, d, e, f, Figure 5b,e, Supplementary Figure 5a,b, Supplementary Figure 6e, f, Figure 7h, Supplementary Figure 7b, c, i**). In addition, most imaging experiments were also confirmed by corresponding SDS-PAGE and western blot analysis. Hence the confidence in our data is very high.

Comment 5: The coherence of the experimental design to the conclusion of the Dsc3 ubiquitin ligase in selecting orphaned proteins at the Golgi for ubiquitination and degradation may also need consideration. For example, what experiments in this work extend the authors' prior study in ref 7 that the ubiquitination is in the Golgi? Are the markers for these Golgi experiments in the current submitted manuscript sufficiently justified for this conclusion? The epifluorescence in ref 7 with markers chosen may be clearer than those in the current manuscript. For the conclusion of a Golgi-located quality control machinery monitoring transmembrane domain length for orphaned membrane proteins, the authors may consider a focused paper on this conclusion.

Answer: Our new experiments (including the new proteomic *vld1Δ/gld1Δ* dataset, **new Supplementary Figure 1, Supplementary Table 1, 2**) show that the Dsc complex patrols the early Golgi to detect and degraded orphaned proteins and thereby prevents their spreading beyond the Golgi. In most cells (93% n=75 cells), the Dsc complex, Ubx3-mNeongreen (Ubx3-NG) partially co-localizes with the early Golgi marker Mnn9, or localizes to its direct vicinity (**new Figure 5e**). In fewer cells (14% n=58 cells), Ubx3-NG was detected at the trans-Golgi with Sec7. Co-localization with the endosomal marker (Vps4) was infrequent. In 91% of cells lacking the Golgi resident Dsc complex (n=82), orphaned proteins (e.g. Hmx1) reached the trans-Golgi (Sec7) (**new Figure 3g**), from where they spread into the endo-lysosomal pathway.

Comment 6: The authors may consider a clearly defined, logical, coherent, experimental strategy and data with n-3 biological replicates and quantification for all experiments.

Answer: We hope that our comments to the reviewer's concerns, and the new experiments have helped to improve the manuscript.

Reviewer #2

Summary of our manuscript: The manuscript submitted to Nature Communications by Teis and colleagues reports first on a characterization of the substrates, effects, and transmembrane domains (TMDs) that are regulated by Endosome and Golgi Associated Degradation (EGAD), a pathway that exhibits several ERAD-like components and features, including an E3 ubiquitin ligase complex. To define the substrates targeted by this ligase, known as Tul1, a proteome-wide screen was conducted using SILAC in wild type and Tul1 delete cells, and the results were compared in an ESCRT-deficient strain, i.e., *vps4*. The efficacy of the Tul1 screen was verified by the identification of ~50% of all TMD proteins as well as a known substrate, Orm2, and amongst the substrates it was striking that 21 were known ER/Golgi proteins and 44% possess short (e.g., <18 aa) TMDs; ~66% had at least one TMD <16 aa. Although there was some overlap in the *tul1* and *vps4* screen, amongst the 72 proteins in the latter screen, they were generally longer. Consistent with the regulation of lipids accorded by Orm2 (and prior work from the Teis lab), PC and PE levels rose in the mutant as well as in a *gld1* mutant, which directs the complex to the Golgi. There was also an apparent increase in short/asymmetric lipids. Interestingly, one new EGAD substrate identified in this study—which is also targeted by ESCRT—is a conserved heme oxidase, Hmx1, and specific assays confirmed combined ubiquitin-proteasome- and ESCRT-dependent degradation. Based on these data, the authors examined Hmx1 residence after beta-estradiol induction: The protein resided in the class E/limiting vacuole membrane and remained at this location after CHX addition in a *tul1vps4* double mutant. A role for Cdc48, along with the Ubx3 UBX domain, in degradation was also clearly shown, and protein retrotranslocation was suggested (but see below). Next, the focus switched to an analysis of Yif1, which was absent from the proteomic screen but is a known Dsc/ESCRT substrate. As anticipated, protein turnover required Tul1, Cdc48, and Ubx3 activities, and as also expected the protein resided in the Golgi. Yet, overexpression freed the protein from the Golgi for ESCRT-dependent vacuolar degradation. The focus of the manuscript next turned to data consistent with the notion that the UBA-containing Dsc2 rhomboid-like iRhom protein is a recognition factor, or more appropriately is at least required for recognition of three substrates: Hmx1, a stabilized form of Orm2, and Yif1 since co-IP analyses in a *dsc2* mutant significantly decreased, to varying extents, association with the Dsc components, and most notably Tul1. Deletion of Ubx3 gave similar results, with the exception of one substrate. Nevertheless, these results are consistent with the function of iRhoms in other systems, along with the importance of Cdc48 recognition via a Ubx protein. Finally, MD simulations were used to suggest that Dsc2, like several other iRhoms, tethers the bilayer, which may facilitate membrane extraction, and the regions within Hmx1—since it contains only a single TMD—that act as a putative 'degron' for Dsc2 were examined. Ultimately, the authors show that adding 1 or 2 additional alpha-helical turns progressively increases the stability of a simplified version

of Hmx1 and that substrate interaction with Dsc2 and Ubx3 drops. Curiously, the Hmx1 substrate used here was preferentially K-48 ubiquitinated.

Overall, several quite intriguing and important results are presented in this study, yet some conclusions are overstated, which could be rectified by the application of more refined technologies. Other results are in line with what might be expected based on studies in related systems. Thus, there are questions about the manuscript's novelty, at least as it currently stands.

Answer: We thank the reviewer for the thorough evaluation of our manuscript and for the insightful, constructive comments. The lengthy revision process was, in part, caused by lab accident that started a fire and devastated our equipment room and by an accident of the first author, who sliced through his middle finger, while cutting up a melon. Therefore, we recruited additional scientists, who are now new co-authors, to support the revision process.

We have addressed the reviewer's concerns with additional experiments that support and extend our original conclusion. Our revised manuscript reports the the following key findings:

- (1) The identification of additional new endogenous substrates for the Dsc complex.
- (2) The finding that these substrates carry short TMD as degrons.
- (3) These degrons can be detected by the Dsc2 subunit, which is required to establish substrate interaction with Dsc complex.
- (4) The Dsc complex initiates K48 or K63 ubiquitination for subsequent degradation via the EGAD (Orm2, Yif1, Hmx1) pathway, or the ESCRT (Hmx1) pathway, or both (Hmx1).
- (5) Dsc complex dependent substrate degradation requires membrane thinning by the L1 loop region of Dsc2
- (6) EGAD (but not ESCRT) dependent degradation requires the UBX domain of Ubx3 for efficient substrate ubiquitination during the membrane extraction process.
- (7) Loss of EGAD leads to the accumulation of orphaned proteins at the Golgi, endosomes and vacuoles, and results in the concomitant upregulation of asymmetric lipids.

We believe that these findings contribute to a novel (and in our view unexpected) mechanistic understanding of the Dsc complex in Golgi quality control, with a key role in targeting orphaned membrane proteins into the EGAD and/or ESCRT pathways for degradation and thereby preventing their spreading from the Golgi to other organelles.

Therefore, our work provides the basis for detailed mechanistic studies on Golgi quality control in the future.

Comment 1: While the authors focus on the 2-fold increase in PI-C-26 species, the effect on PE in Fig 2 may be just as profound. Is there a statistical difference between the effects on PI and PE? Is the same conclusion true for PC?

Answer: Yes, thank you for pointing this out! Also, asymmetric C-26 PA, PE, and PC species double. We focused on PI-C26 species because asymmetric PI reaches almost 8 mol% of PI in *tull1Δ* and *gld1Δ*, whereas for asymmetric PA, PC, PE the levels amount to < 2 mol% of the respective classes (**new Figure 2e**).

Comment 2: Is the integrated FLAG-Hmx1 protein fully active, as determined in growth assays for the function of this protein?

Answer: Untagged Hmx1 was identified in three unbiased Mass-Spec experiments as a Dsc complex substrate, and different N-terminally tagged versions (GFP, FLAG, ALFA) are turned over just like the untagged Hmx1. Based on these results, we assume that the tagged versions of Hmx1 recapitulate the degradation pathways of the untagged protein. Also, N-terminal tagging of Hmx1 with GFP, Flag or ALFA, did not affect cell growth (Figure 1 for the reviewer). Therefore, it is not likely N-terminal tagging of Hmx1 generates a toxic protein.

We went at great length to further test the functionality of tagged Hmx1, and devised different growth assays that have been used earlier to examine how loss of Hmx1 affects (1) the cellular capacity to utilize heme as iron source (in *fet3Δ hmx1Δ* double mutants) (10.1074/jbc.M306584200) or (2) the response to oxidative stress (H₂O₂) (10.1021/bi061429r.). Under all conditions tested, the *fet3Δ hmx1Δ* mutant complemented with Flag-Hmx1 or GFP-Hmx1 grew comparable to the *fet3Δ hmx1Δ* complemented with untagged Hmx1. Similarly, the sensitivity of an *hmx1Δ* strain complemented with Flag-Hmx1 or GFP-Hmx1 to H₂O₂ was comparable to the *hmx1Δ* complemented with untagged Hmx1. In contrast to these earlier reports (10.1074/jbc.M306584200; 10.1021/bi061429r.), we did not detect growth defects for the *fet3Δ hmx1Δ* or the *hmx1Δ* under these conditions (Figure 1 - provided for the reviewer, showing growth of the indicated strains in iron free media with or without hemin).

Figure 1. Loss of Hmx1, untagged Hmx1, Flag-Hmx1 or GFP-Hmx1 do not inhibit cell growth in iron free media.

Comment 3: The statement, “loss of ESCRT function did not block Hmx1 degradation...” (p. 7) should be altered to the “loss of ESCRT function [alone] did not block...”. In addition, this phenomenon is not uncommon, and is seen for ERAD substrates that escape the ER and are degraded in an ESCRT-dependent fashion.

Answer: Thank you for pointing this out. We have included the following statements and the corresponding citations on **page 7 and page13:**

page7, line 264: Since loss of ESCRT function alone did not block Hmx1 degradation....

page 14, line 504: ...Perhaps conceptually related is the phenomenon that substrates that escape ERAD, are degraded in an ESCRT-dependent fashion ([10.1091/mbc.e09-10-0910](https://doi.org/10.1091/mbc.e09-10-0910), [10.1074/jbc.M111.233346](https://doi.org/10.1074/jbc.M111.233346)). It is currently unknown if Dsc complex contributes to the detection of ERAD escapers and targets such proteins for degradation.

Comment 4: The results with the K63 and K48 nanobodies are intriguing, but confidence for the authors' conclusions would be strengthened if the resulting elutes were blotted for antibodies specific for K48 and K63. Moreover, since mixed chains with K11 have been uncovered as being preferentially targeted for proteasome-dependent degradation, an analysis of this species should also be performed.

Answer: New results (**new Figure 4c, Supplementary 4c**) confirm that the Dsc complex was required to initiate mainly K48 and K63 ubiquitination of Hmx1.

(1) We have used yeast strains in which endogenous ubiquitin is replaced with either K11R or K48R or K63R ubiquitin (the so-called SUB strains [10.1128/MCB.15.3.1265](https://doi.org/10.1128/MCB.15.3.1265)). Performing denaturing immunoprecipitation (IP) of Hmx1 from these strains, showed that loss of K48 polyubiquitin chains, almost completely abrogated Hmx1 polyubiquitination (**new Supplementary Figure 4c lanes 5, 6**). In cells that cannot form K63 polyubiquitin chains, a Hmx1 polyubiquitination was reduced, but a significant fraction of Hmx1 remained (probably K48) polyubiquitinated (**new Supplementary Figure 4c, lanes 3, 4**). Loss of K11 polyubiquitin chain formation does not seem to reduce overall Hmx1 polyubiquitination (**new Supplementary Figure 4c, lanes 7, 8**).

(2) The K48 linkage specific antibody also showed that immunoprecipitated Hmx1 was K48 polyubiquitinated, which required the Dsc complex (**new Figure 4c, lanes 1, 2**). In our hands, two different K63 linkage specific antibodies (from Merck/Sigma Aldrich and Cell signaling) still showed residual laddering in the K63R strain, therefore we have not used it for further analysis (data not shown).

Comment 5: The quality of the blot shown in Fig 4D is poor. Replicates of this important experiment should be shown, along with simple quantification of whole band densities. It is also expected that ubiquitinated Hmx1 should be preferentially extracted into the S100 fraction. Is this true?

Answer: Done. The new experiments and the quantification (**new Figure 4d, e**) confirm our original observation. Only if Cdc48 is active, ubiquitinated Hmx1 is detected in the soluble fraction, where it accumulates if proteasomal degradation is blocked. Of note, due to the

improved experimental setup, we now also detected changes in the ubiquitination pattern of Hmx1, once Cdc48 function was blocked. Similar changes in Hmx1 ubiquitination were observed with the Ubx3^{ΔUBX} mutant (**new Figure 4h**).

Comment 6: The importance of the vps4-EQ mutant is not discussed in the text (Fig. 4F), and data to fully support the statement that, “K48 polyubiquitination [is needed] for EGAD” are incomplete.

Answer: We have edited the text accordingly:

page 9, line 322: ‘Yet, when the ESCRT pathway was also disrupted in *ubx3*^{ΔUBX} cells, (either by expressing the dominant negative of Vps4^{EQ} mutant or in *vps4Δ* cells), EGAD mediated Hmx1 degradation was hampered’

Vps4^{EQ} is introduced earlier in the manuscript on **page 8, line 292:** ...in cells that expressed dominant negative Vps4^{E233Q} (hereafter Vps4^{EQ}) to disrupt the ESCRT pathway....

We provide new data with K48 linkage specific antibodies and SUB strains (**new Figure 4c, Supplementary Figure 4c**) showing that in ESCRT deficient cells, Hmx1 is K48 polyubiquitinated and that the Dsc complex is required for K48 ubiquitination of Hmx1 and EGAD.

Comment 7: The logic underlying the tangential focus on Yif1 is unclear. Does the protein have short TMDs? Amongst other Dsc/ESCRT substrates, are there unique features that make Yif1 particularly worthy of study? If not, the data in Fig 5 could be included as a supplement and referred to in the context of the substrate recognition section.

Answer: Yif1 does indeed have a short TMD1 (18 amino acids, please see discussion on page 14). Yet, given the comments of reviewer 1, we have substantially shortened the section on Yif1, and now only provide the data showing that endogenous Yif1 interacts with the Dsc complex (**new Figure 5a, b, Supplementary Figure 5c**) and that it is degraded in proteasome by EGAD (**Supplementary Figure 5a, b**).

Comment 8: The results in Fig 6 need to be repeated for statistical validation. Does removal of the UBX domain in Ubx3 give a null phenotype in this assay? Do we know which regions in Dsc2 are required specifically for ‘recognition’? Does the loss of Dsc2 prevent retrotranslocation (viz Fig 4D)? While the results are not unexpected, more refined tools (such as site-specific crosslinking, which could show direct recognition of the TMD) might excitingly establish that Dsc2 is the prime recognition factor. This result that would substantially increase the novelty of the study.

Answer: The results in Figure 6 have been repeated and quantified (please see **new Figure 5a, b**).

To address if ‘removal of the UBX domain in Ubx3 give a null phenotype in this assay’, we have conducted additional experiments. Our new results (**new Figure 4h**) show that the UBX domain plays important roles in controlling Hmx1 ubiquitination for EGAD mediated proteasomal degradation. Loss of the UBX domain impaired ubiquitination of Hmx1 in the membrane, hampered efficient extraction and proteasomal degradation. Consistently, the deletion of the UBX domain in Ubx3 impaired (but did not fully block) EGAD mediated degradation Hmx1 and Orm2 degradation, while the UBX domain was not required for ESCRT dependent degradation of Hmx1 (**Figure 4f, g, Supplementary Figure 4d-g**).

In the *dsc2Δ* mutant, Hmx1 is not ubiquitinated (**Figure 3b**), Hmx1, Orm2, and Yif1 cannot interact with the Dsc complex (**new Figure 5a, b, Supplementary figure 5c**). Hence in *dsc2Δ* mutants, but also membrane thinning deficient *Dsc2^{Y53A,R54A}* (**new Figure 6a-c**) mutants, these proteins are no longer degraded and accumulate on post-ER organelles.

As suggested by the reviewer, that have used refined approaches to identify regions in Dsc2 that interact specifically with the TMD of Hmx1. Yet, despite extensive efforts, using MD simulations of Dsc2 and with the TMD of Hmx1 in a lipid bilayer (Figure 2a – for the reviewer) in combination with experimental approaches, **we failed to identify residues in Dsc2 that specifically cross-linked with the TMD of Hmx1**. We believe that our efforts were hampered (1) by technical difficulties caused by extensive cross-links between the TMD deignons (T292C, H291C) (Figure 2b for the reviewer), which probably hindered Dsc2-TMD crosslinks or (2) by inaccurate modeling / docking of the TMD with Dsc2 (see Figure 2a for reviewer).

Figure 2. **a** Snapshot of an MD simulation with Dsc2 and the TMD of Hmx1 (Hmx1²⁸⁴). The residue shown as sticks were mutated to cysteine and used for cysteine specific cross-linking. **b** On bead zero length cross-linking with Cu²⁺ phenanthroline on ALFA-Hmx1²⁸⁴ immunoprecipitated complexes.

Interestingly, in the course of these experiments, we identified specific cross-links between Ubx3-Dsc2 Q238C bound to the TMD of Hmx1 (Figure 2b, lanes 5, 7, 8, provided for the reviewer).

Using the same experimental setup (mutational analysis in combination with MD simulations), we now show that local membrane thinning by L1 loop region (Y53, R54) in Dsc2 is required for Orm2 and Hmx1 degradation (**new Figure 6, Supplementary figure 6a, e, f**).

Other new experiments (**new Figure 5a, b, Supplementary figure 5c, new Figure 7g**) indicate that Dsc2 is required for interaction with the TMD degnon of Hmx1. To arrive at this conclusion, we performed additional reciprocal co-immunoprecipitation experiments (**new Figure 5a, b, Supplementary figure 5c**), which show that Dsc2 is required for the interaction of the Dsc complex with the substrates. In addition, we have strongly overexpressed the TMD of Hmx1 together with Dsc2 in *dsc3Δ* cells. Overexpressed Dsc2 which is present in vast excess, co-purified efficiently with the TMD of Hmx1, indicating that Dsc2 alone can interact with the TMD of Hmx1. Collectively, these experiments support the hypothesis that Dsc2 has the capacity to engage the TMD degnon of Hmx1. Furthermore, we now also show this TMD of Hmx1 is a transposable degnon (**new Figure 7h, Supplementary Figure 7i**)

Collectively we believe that these results support the idea that Dsc2 detects short TMD degnons to establish interaction with the Dsc complex for ubiquitination and EGAD or ESCRT mediated degradation.

Yet, it is also clear that more work is needed in the future to characterize mechanistically how Dsc2 detects TMD degnons. These additional experiments will require *in vitro* reconstitution experiments and go beyond the scope of our current work, given the breath of our study already.

Comment 9: The data in Fig S6 suggest variable Tull1 loading for several of the strains. This emphasizes the need for replicates of the co-IP experiments and necessitates a quantification of the relative amount of protein precipitated (divided by the input) in the various backgrounds used.

Answer: To directly compare the interaction of Orm2, Hmx1 and Yif1 with the Dsc complex and for quantification, we used additional reciprocal immunoprecipitation experiments (**new Figure 5a, b**). Immunoprecipitation of Ubx3-Flag from WT cells, *dsc2Δ*, *dsc3Δ*, and *tull1Δ* cells confirmed that Dsc2 is key for the interaction of the Dsc complex with its endogenous substrates Orm2, Yif1 and Hmx1. These experiments have been quantified.

We apologize for the poor quality of the α -Tull1 anti-serum in the input sample, which is caused by strong unspecific bands. However, we have confidence in the antibody, since the antibody works well in the immunoprecipitated samples and the specific bands are absent in the *tull1Δ* cells (**Figure 5a, Supplementary Figure 5c**). The reduction in Tull1 protein levels in the

different *dsc* mutants is a known (yet little understood) biological effect, consistent with earlier observations (10.1074/mcp.M114.040774). It will be interesting to explore this phenomenon in the future, but for the interaction studies here, it does not matter, since Tull1 is not required for substrate interaction.

Comment 10: Studies to identify the Hmx1 degron for Dsc2 in Fig 7 require confidence that Dsc2 is indeed the direct recognition factor (see above).

Answer: Please see answers to comment 8 and to comment 13.

Comment 11: In Fig 7E, do the levels of Tull1 interaction also decline?

Answer: In this experiment, the model substrates were immunoprecipitated from *tull1Δ* cells to compare the interaction of the different model substrates with the Dsc complex (**now Figure 7c, new Supplementary Figure 7c**). By performing the experiments in *tull1Δ* strains, we ensure that the levels of the different model substrates are similar and thus suited to compare the interaction with the Dsc complex. Would we have used WT cells, the steady state protein levels of the model substrates with longer TMDs would be substantially higher (Supplementary Figure 7a), which would make it difficult to compare their interaction with the Dsc complex.

Comment 12: In the absence of a live-cell ER marker, it is unclear if the protein localizes to the ER based simply on the images shown in Fig 7G and in Fig S7G.

Answer: In a set of new experiments (**new Figure 3g, Supplementary Figure 3d**), we have used Elo3 as an ER marker, and Sec7 as a late Golgi marker and followed the trafficking of GFP-Hmx1. These new results demonstrate that in cells lacking the Dsc complex, full length Hmx1 is first at the ER, and then accumulates outside the ER, where it partially co-localizes with the late Golgi marker Sec7, from where it spreads into the endo-lysosomal pathway (**Figure 3f, new Figures 3g, Supplementary Figure 3d**).

We did not directly co-localize the model substrate GFP-Hmx1²⁶⁴ with Elo3-mCherry, since it behaves similar to the full length Hmx1 in most experiments. Yet we believe that the new experiments with full length GFP-Hmx1 (**new Figures 3g, Supplementary Figure 3d**), in combination with the images of the model substrate GFP- or phluorin-Hmx1²⁶⁴ (**Figure 7d, Supplementary Figure 7h, Supplementary Figure 7e**) show that vacuoles / class E compartments, labeled with FM4 or mCherry-Cps1, are distinct from large Hmx1²⁶⁴ containing structures, that are reminiscent of nuclear ER (e.g. see in dividing cells). Yet, based on the comments of the reviewer, we have toned down the argument accordingly.

Comment 13: The definition of a degron requires that a motif is completely transposable to another protein substrate and that the requirements for degradation are maintained. It does not appear that this condition was met.

Answer: Thank you for this suggestion. We have added new experiments, that show that the TMD degron of Hmx1 is transposable (**new Figure 7h, Supplementary Figure 7i**). Towards this goal we have generated a chimeric protein, consisting of the cytosolic portion of Pep12 (an endosomal tail anchored t-SNARE) and the TMD (aa 284-317) of Hmx1. The Pep12-Hmx1-TMD chimera became a substrate of the Dsc complex that was efficiently targeted for vacuolar degradation (**new Figure 7h, Supplementary Figure 7i**).

Response to referees

Comments Reviewer #1:

The authors report SILAC labelling proteomics to report proteins in wt, *tul1Δ*, *gld1Δ*, *vld1Δ* cells. Proteins increasing in abundance in the gene deleted cells were reported with predicted transmembrane lengths. Lipidomics is reported for wt, *tul1Δ*, *gld1Δ*, *Orm2KR* mutant cells.

Expressed FLAG tagged Hmx1 IP was concluded to be polyubiquitinated based laddering in western blots with cycloheximide addition to cells expressing FLAG-Hmx1 concluded to lead to degradation but not in *tul1Δ*, *dsc2Δ*, *ubx3Δ*, *dsc3Δ* or *gld1Δ* cells. The effect of MG-132 is reported in wt and *vps4Δ* cells. Replacing Hmx1 with β -estradiol inducible mNeonGreen-ALFA-Hmx1 is reported with imaging for wt cells (replacing Hmx1) *tul1Δ* cells, *tul1Δ* *rsp5G747E* or *rsp5G747E* cells. Replacing Hmx1 with β -estradiol inducible mNeonGreen-ALFA-Hmx1 and expressing mCherry-Cps1 is reported as well in *vps4Δ* cells and *tul1Δvps4Δ* cells.

FLAG-Hmx1 expressing wt, or *vps4Δ*, or *tul1Δvps4Δ*, or *tul1Δ*, or *Vps4EQ* and *Ub-K63R*, or *Ub-K48R* or *Ub-K11R* cells were assessed with antiK48ubiquitin antibodies and nanobodies. FLAG-Hmx1 IP from S100 or P100 fractions from *vps4Δ* or *vps4Δ cdc48-3* cells with or without MG-132 is reported as well as *ubx3ΔUBX* cells and *Vps4EQ* mutant and *vps4D* cells and β -estradiol241inducible mNeonGreen-ALFA-Hmx1 cells.

The authors report IP studies with *Orm2*, *Yif1* in FLAG-Hmx1 expressing cells deleted or not for indicated genes as well as cells expressing *Ubx3-FLAG* as well as imaging of wt and gene deleted cells expressing *Ubx3-Neon Green* and *dsRed-HDEL*, *Vps4-mCHERRY*, *Sec7-mCHERRY*, *Mnn9-BFP*.

Molecular dynamic simulations of the rhomboid like domain of *Dsc2* or *Dsc2Y53A,R54A* in 45% POPC, 45% POPE and 10% cholesterol with CHARMM-GUI simulated with indicated lipids is reported. Experiments with mutant *Dsc2Y53A,R54A* are reported in cells for GFP-*Orm2* and *NeonGreen-Hmx1* imaging.

Expression of indicated Hmx1 mutant constructs N-terminally fused to mNeonGreen-ALFA-(NG)-tags or GFP-ALFA-(GFP)-tags with western blots of cells with beta-estradiol or cycloheximide. Imaging of expressed GFP- Hmx1 is indicated and denaturing and non-denaturing IP.

The authors conclude a Dsc quality control at the Golgi for orphaned proteins with short transmembrane domains that they also conclude are degrons. *Dsc2* thinning of membranes is reported with associated *Ubx3* enabling *cdc48* extraction to proteasome degradation of the selected orphaned proteins and *Tul1* ubiquitination concluded from the studies.

Again, this reviewer could not find HMX1 in supplementary Table 1 tab WT-*tul1d* although it is in the other 2 tabs, wt-*vps4d*, *vld1d-gld1d*. This makes it confusing for Figure 1b for example. There is no mention of the SILAC proteomics conclusions in the Discussion that remain focussed on the proteins indicated in Figure 7i with HMX1 absent from supplementary Table 1 tab wt-*tul1d*. The authors have also indicated that *Yif1* is absent from the SILAC proteomics. The paper remains an extension of their prior studies on HMX1 and *Yif1* and if focussed on the claims of the discussion without the proteomics may be more coherent. Again for the

lipidomics, did this lead directly to the design of the molecular dynamic simulations? Is the lipidomics needed or of archival value?

An argument can be made for Figure 2 but the missing HMX1 in supplementary table 1 remains confusing. The figures have been improved. A minor point is that the authors may consider SD rather than SEM for the figures. In Supplementary Information there does not seem to be any legends to describe the 3 supplementary movies.

Response:

We thank the reviewer for the careful analysis of our revised manuscript.

To understand how the Dsc complex detected and degraded its substrates, we focused on Hmx1, because it was among the top 5 ranking Gld1-Dsc complex potential membrane protein substrates that were identified in our proteomic analysis (Figure 1, Supplementary Data 2, highlighted in red).

Hmx1 can be found in Supplementary Table 1 (now called Supplementary Data 1) in tab1 line 9; tab2 line 11 and tab3 line 24 and in Supplementary Table 2 (now called Supplementary Data 2) in tab1 line 5; tab2 line 6; and tab3 line 17. In addition, we have highlighted Hmx1 in red.

We provide legends for the Supplementary Movies 1-3.

Comments Reviewer #2 (Remarks to the Author):

The authors have satisfactorily addressed the concerns of this reviewer.

Minor issues:

1. The 3rd column in Fig 5e is not labeled. Is this "dsRED-HDEL"?
2. Fig 7g legend doesn't refer to "dsc3delta cells", nor does the figure, which is what is what appears to be stated in the text.

Response:

We thank the reviewer for the positive feedback and for the careful analysis of our revised manuscript.

We have addressed these minor issues. Therefore we have labeled the 3rd column of Figure 5e and we have added '*dsc3Δ*' in the Figure 7g.